# Opposite polarity programs regulate asymmetric subsidiary cell divisions in grasses

**Dan Zhang[1], Roxane P Spiegelhalder[2], Emily B Abrash[3], Tiago DG Nunes[1], Inés Hidalgo[1], M Ximena Anleu Gil[3], Barbara Jesenofsky[1], Heike Lindner[1,2], Dominique C Bergmann[3,4]*, Michael T Raissig[1,2]***

[1]Centre for Organismal Studies Heidelberg, Heidelberg University, Heidelberg, Germany; [2]Institute of Plant Sciences, University of Bern, Bern, Switzerland; [3]Department of Biology, Stanford University, Stanford, United States; [4]Howard Hughes Medical Institute, Stanford University, Stanford, United States

**Abstract** Grass stomata recruit lateral subsidiary cells (SCs), which are key to the unique stomatal morphology and the efficient plant-atmosphere gas exchange in grasses. Subsidiary mother cells (SMCs) strongly polarise before an asymmetric division forms a SC. Yet apart from a proximal polarity module that includes PANGLOSS1 (PAN1) and guides nuclear migration, little is known regarding the developmental processes that form SCs. Here, we used comparative transcriptomics of developing wild-type and SC-less *bdmute* leaves in the genetic model grass *Brachypodium distachyon* to identify novel factors involved in SC formation. This approach revealed BdPOLAR, which forms a novel, distal polarity domain in SMCs that is opposite to the proximal PAN1 domain. Both polarity domains are required for the formative SC division yet exhibit various roles in guiding pre-mitotic nuclear migration and SMC division plane orientation, respectively. Nonetheless, the domains are linked as the proximal domain controls polarisation of the distal domain. In summary, we identified two opposing polarity domains that coordinate the SC division, a process crucial for grass stomatal physiology.

*For correspondence: dbergmann@stanford.edu (DCB); michael.raissig@ips.unibe.ch (MTR)

## Editor's evaluation

This study presents a landmark finding that two proteins previously implicated in plant cell polarity, but in different cell types and species, function cooperatively to polarize an asymmetric division in the grass Brachypodium distachyon. The results are well-documented and convincing. The work will be of interest to cell and developmental biologists generally and also to those broadly interested in plant biology.

## Introduction

Stomata are 'breathing pores' in the plant epidermis that regulate gas exchange between the plant and the environment. They open to enable the entry of carbon dioxide to be assimilated in photosynthesis and close to restrict water vapour loss. Efficiently balancing carbon dioxide uptake and water vapour loss is key for plant survival and abiotic stress resilience. Grasses like *Zea mays* (maize), *Oryza sativa* (rice), and the genetic model system *Brachypodium distachyon* present a unique stomatal morphology, where a pair of central, dumbbell-shaped guard cells (GCs) is flanked by two lateral subsidiary cells (SCs) (*Nunes et al., 2020*; *Stebbins and Shah, 1960*). The grasses' stomatal morphology is linked to faster stomatal movements that contribute to more water-use efficient gas exchange (*Franks*

**Figure 1.** *BdPOLAR* is a novel regulator of subsidiary cell (SC) division in the *Brachypodium distachyon* stomatal lineage. (**A**) Stomatal development in *B. distachyon*; in a stomatal file (1) (grey asterisk) a transverse asymmetric division (2) generates a guard mother cell (GMC; blue, with blue asterisk), which laterally induces subsidiary mother cells (SMCs; yellow) (3). SMCs divide asymmetrically (4) before GMCs divide symmetrically (5), and the complex matures (6). (**B**) Volcano plot displaying wild-type (WT) expression level (y-axis) and fold change in *bdmute* compared to WT (x-axis) of all expressed genes in the developmental zone (dots = genes). Green, large dots indicate genes significantly downregulated in *bdmute* and purple, large dots indicate genes significantly upregulated in *bdmute*. The red circle indicates *BdPOLAR*. (**C**) Differential interference contrast images of the epidermis in WT (Bd21-3), *bdpolar-1*, *bdpan1-1,* and *bdpolar-1; bdpan1-1* (third leaf, 19 days after germination [dag]). Aberrant SCs are indicated with white asterisks. Scale bar, 15 µm. (**D**) Quantifications of defective SCs of segregating lines after crossing (*bdpolar-1;+/+, bdpolar-1;bdpan1-1/+,* and *bdpolar-1;bdpan1-1*), the parental lines *bdpolar-1* and *bdpan1-1*, and a WT control. Samples were compared using a one-way ANOVA and post hoc Tukey test for multiple comparisons; different letters indicate significant differences ($p<0.05$). n=6–9 individuals and 643–848 SCs per genotype.

The online version of this article includes the following source data and figure supplement(s) for figure 1:

**Source data 1.** Quantifications of defective SCs in *bdpolar-1, bdpan1-1,* and *bdpolar-1;bdpan1-1*.

**Figure supplement 1.** CRISPR/Cas9 and EMS (ethyl methanesulfonate)-mutagenised mutants in *BdPOLAR* and *BdPAN1*.

**Figure supplement 1—source data 1.** Quantifications of defective subsidiary cells in *bdpolar-1, bdpolar-2,* and *bdpolar-3*.

**Figure supplement 2.** Misoriented subsidiary mother cell (SMC) division planes likely cause aberrant subsidiary cells in mature leaf zones.

**Figure supplement 2—source data 1.** Original, unmodified and uncropped gel image.

*and Farquhar, 2007*; *Lawson and Vialet-Chabrand, 2019*; *McAusland et al., 2016*). Among other features, the grasses' morphological innovations to stomatal form and function likely contributed to their evolutionary success (*Hetherington and Woodward, 2003*) and their importance in modern agriculture (*FAO Statistical Pocketbook FAO, 2015*).

The linear epidermal cell files in a growing grass leaf follow a strict and stereotyped developmental gradient from base to tip (*Figure 1A*; *Stebbins and Shah, 1960*). Some cell files acquire stomatal identity at the leaf base, where a transversal, asymmetric cell division (ACD) generates a small guard mother cell (GMC) and a large, interstomatal pavement cell precursor (*Figure 1A*; *Raissig et al., 2016*; *Stebbins and Shah, 1960*; *Wu et al., 2019*). The GMC then induces its lateral neighbour cells to become subsidiary mother cells (SMCs), which strongly polarise towards the GMC and undergo an extreme longitudinal ACD to generate a proximal SC and a distal pavement cell (*Figure 1A*; *Cartwright et al., 2009*; *Facette et al., 2015*; *Gray et al., 2020*; *Humphries et al., 2011*; *Nan et al., 2021*; *Raissig et al., 2017*; *Stebbins and Shah, 1960*; *Zhang et al., 2012*). Finally, the GMCs undergo a symmetric longitudinal cell division generating two GC precursors before complex morphogenetic processes shape the so-called graminoid stomatal morphology (*Figure 1A*; *Galatis and Apostolakos, 2004*; *Spiegelhalder and Raissig, 2021*; *Stebbins and Shah, 1960*).

The lateral SCs represent one of two key morphological innovations of grass stomata (*Gray et al., 2020*; *Nunes et al., 2020*). It was shown that SC presence is critical for stomata to open and close quickly in response to environmental cues in grasses (*Raissig et al., 2017*), yet few factors required for SC formation have been identified. SC formation can be divided into three steps: (1) SMC establishment, (2) SMC polarisation, and (3) the ACD that yields the SC. Previous work showed that orthologues of the *Arabidopsis* stomatal bHLH transcription factor *MUTE* are required to establish SMC identity in the three grass species *B. distachyon,* maize, and rice, and that *mute* mutants lack SCs (*Raissig et al., 2017*; *Wang et al., 2019*; *Wu et al., 2019*). In maize SMCs, a proximal polarity module localised at the GMC/SMC interface establishes pre-mitotic polarity and guides the migrating nucleus towards the GMC/SMC interface. ZmBRICK1 (ZmBRK1), a subunit of the SCAR/WAVE regulatory complex (WRC), initially localises at the interface of GMC/SMC (*Facette et al., 2015*). ZmBRK1 then promotes the polarised accumulation of two leucine-rich-repeat receptor-like kinases, ZmPANGLOSS1 (ZmPAN1) and ZmPAN2 (*Cartwright et al., 2009*; *Facette et al., 2015*; *Zhang et al., 2012*). ZmPAN2 acts upstream of ZmPAN1, which then recruits two Rho of plants (ROP) GTPases, ZmROP2 and ZmROP9 (*Humphries et al., 2011*; *Zhang et al., 2012*). The ZmROPs physically interact with and possibly activate WRC components, which lead to actin patch formation at the GMC/SMC interface and nuclear migration (*Facette et al., 2015*). Upon nuclear migration, the phosphatases DISCORDIA1 (ZmDCD1) and ALTERNATIVE DCD1 (ZmADD1) control the formation and positioning of the preprophase band (PPB) (*Wright et al., 2009*), which is a transient cytoskeletal array that guides the future division plane (*Facette et al., 2019*; *Livanos and Müller, 2019*). After the disappearance of the PPB, factors such as the microtubule-binding protein TANGLED1 (TAN1) maintain the cortical division site (*Cleary and Smith, 1998*; *Martinez et al., 2017*). Finally, the myosin XI protein OPAQUE1 (O1)/DCD2 is critical for phragmoplast guidance during late cytokinesis and subsequent cell plate formation to complete the ACD that forms SCs (*Nan et al., 2021*).

In the *Arabidopsis* stomatal lineage, cell polarity and cytoskeleton-mediated nuclear migration and ACDs are regulated by a different set of polarity genes (*Guo et al., 2021b*; *Muroyama and Bergmann, 2019*; *Ramalho et al., 2022*). Breaking of asymmetry in the stomatal lineage (BASL), a dicot-specific protein, serves as an intrinsic polarity factor to regulate ACDs of meristemoid mother cells and forms a polarity crescent distal to the division plane (*Dong et al., 2009*; *Nir et al., 2022*). BASL promotes polarisation of BREVIS RADIX (BRX) family proteins and POLAR LOCALIZATION DURING ASYMMETRIC DIVISION AND REDISTRIBUTION (POLAR) within its own polarity domain (*Pillitteri et al., 2011*; *Rowe et al., 2019*). The *Arabidopsis* polarity proteins can scaffold kinases to regulate cell division capacity, fate asymmetry, and nuclear migration (*Guo et al., 2021b*; *Muroyama and Bergmann, 2019*). POLAR recruits the glycogen synthase kinase 3 (GSK3)-like kinase BIN2 to regulate cell division capacity (*Houbaert et al., 2018*), whereas a positive-feedback loop between BASL and the mitogen-activated protein kinase pathway is essential for enforcing different cell fates after ACD (*Zhang et al., 2015*). BASL and its polarity domain also act to orient nuclear migrations during ACDs (*Muroyama et al., 2020*).

Here, we leveraged the unique *bdmute* phenotype to transcriptionally profile developing leaves that form SCs (wild type [WT]) and leaves that lack SCs (*bdmute*). Among the genes dependent on *BdMUTE,* we identified the homologue of *AtPOLAR, BdPOLAR,* and showed it plays a role in SC formation. BdPOLAR not only occupied a distal polarity domain that is opposite of the proximal PAN1 polarity domain, but also BdPOLAR's polarisation was dependent on *BdPAN1*. Detailed and

quantitative analysis of the *bdpan1* and *bdpolar* phenotypes suggested diverse roles in regulating pre-mitotic nuclear polarisation and division plane orientation, respectively.

## Results

### Comparative transcriptomics identify a role for *BdPOLAR* in SC development

To identify novel factors regulating SC development in the grass stomatal lineage, we took advantage of the fact that loss of *MUTE* in *B. distachyon* did not result in seedling lethality. In rice and maize, *mute* mutants completely abort stomatal development and die, whereas *bdmute* plants are viable and form two-celled stomata that lack SCs (*Raissig et al., 2017*; *Wang et al., 2019*; *Wu et al., 2019*). We profiled the transcriptome of the developing *B. distachyon* leaf zone in WT (with SCs) and in *bdmute* (without SCs; *Figure 1B*). Among the 35 downregulated genes in *bdmute* (*Supplementary file 1*), the fourth most downregulated gene was the homologue of *Arabidopsis POLAR* (At4g31805), a polarly localised protein of unknown function with a role in regulating ACDs in the *Arabidopsis* stomatal lineage (*Houbaert et al., 2018*; *Pillitteri et al., 2011*). To assess the functional relevance of *BdPOLAR* (BdiBd21-3.3G0715200 - Bradi3g54060), we induced targeted mutations at three different positions in the coding sequence using CRISPR/Cas9 (*Figure 1—figure supplement 1C, E, F*). The three alleles *bdpolar-1* (+T), *bdpolar-2* (+T), and *bdpolar-3* (+T) all failed to correctly divide 25–40% of their SCs (*Figure 1—figure supplement 1A, B*). They showed oblique and misoriented SC division planes that resulted in missing or improperly formed SCs in mature leaf zones (*Figure 1C and D*; *Figure 1—figure supplement 1A, B*; *Figure 1—figure supplement 2A, B*). These defects in asymmetrically dividing SMCs suggested that *BdPOLAR* played a role in regulating SMC polarisation and/or the formative SC division itself and that, unlike in *Arabidopsis* (*Houbaert et al., 2018*), a single mutation affecting the *BdPOLAR* homologue severely affected grass stomatal development.

### *BdPOLAR* and *BdPAN1* are both required during the formative SC division

The *bdpolar* division defects were reminiscent of classical maize SMC polarity mutants like *zmpan1* and *zmpan2*, which caused incorrectly positioned, oblique ACDs in SMCs (*Cartwright et al., 2009*; *Facette et al., 2015*; *Zhang et al., 2012*). A *BdPAN1* mutant (BdiBd21-3.3G0526300 - Bradi3g39910, *bdpan1-1*) isolated in a screen for stomatal pattern defects contained a nine-base pair (bp) in-frame deletion near the 5′-end of *BdPAN1* and displayed ~45% aberrant and oblique SC divisions (*Figure 1C and D*; *Figure 1—figure supplement 1D, F*). This suggested that *bdpan1* and *bdpolar* mutants caused similar phenotypes, albeit with different penetrance.

Examination of the developmental origins of defective SCs revealed that early transverse divisions generating GMCs were normal in both *bdpolar-1* and *bdpan1-1*. The longitudinal ACDs in SMCs, however, showed misoriented divisions. In both *bdpolar-1* and *bdpan1-1* some of the cell division planes were transverse rather than longitudinal (*Figure 1—figure supplement 2A, B, C*). This suggested that the origin of the aberrant SCs was incorrectly oriented ACDs during SC recruitment.

To explore the functional relationship between *BdPOLAR* and *BdPAN1*, we crossed the single mutants to generate the *bdpolar-1;bdpan1-1* double mutant. The double mutant displayed a much stronger phenotype with ~82% of abnormally divided SCs (*Figure 1C and D*). Accordingly, the misoriented division planes in SMCs were overly abundant in *bdpolar-1;bdpan1-1* (*Figure 1—figure supplement 2D*). This suggested that *BdPOLAR* and *BdPAN1* either work redundantly in the same cellular process or that they affect different elements of a complex pathway to properly orient and execute SMC divisions. However, neither the single mutants nor the double mutant showed any growth or fertility defects in well-watered and optimised growth conditions.

### The polarity domains of BdPOLAR and BdPAN1 are mutually exclusive

The observed SMC division orientation phenotypes and the behaviour of *BdPAN1* and *BdPOLAR* homologues in other species suggested that these proteins might be polarly localised in GMCs and/ or SMCs. A *BdPAN1p:BdPAN1-YFP* reporter was expressed in all young protodermal cells, and the protein was evenly distributed at the cell periphery (*Figure 2A*; *Figure 2—figure supplement 1A*). Once GMCs were formed and started to elongate (stage 3; *Figure 1A*), BdPAN1 protein began to

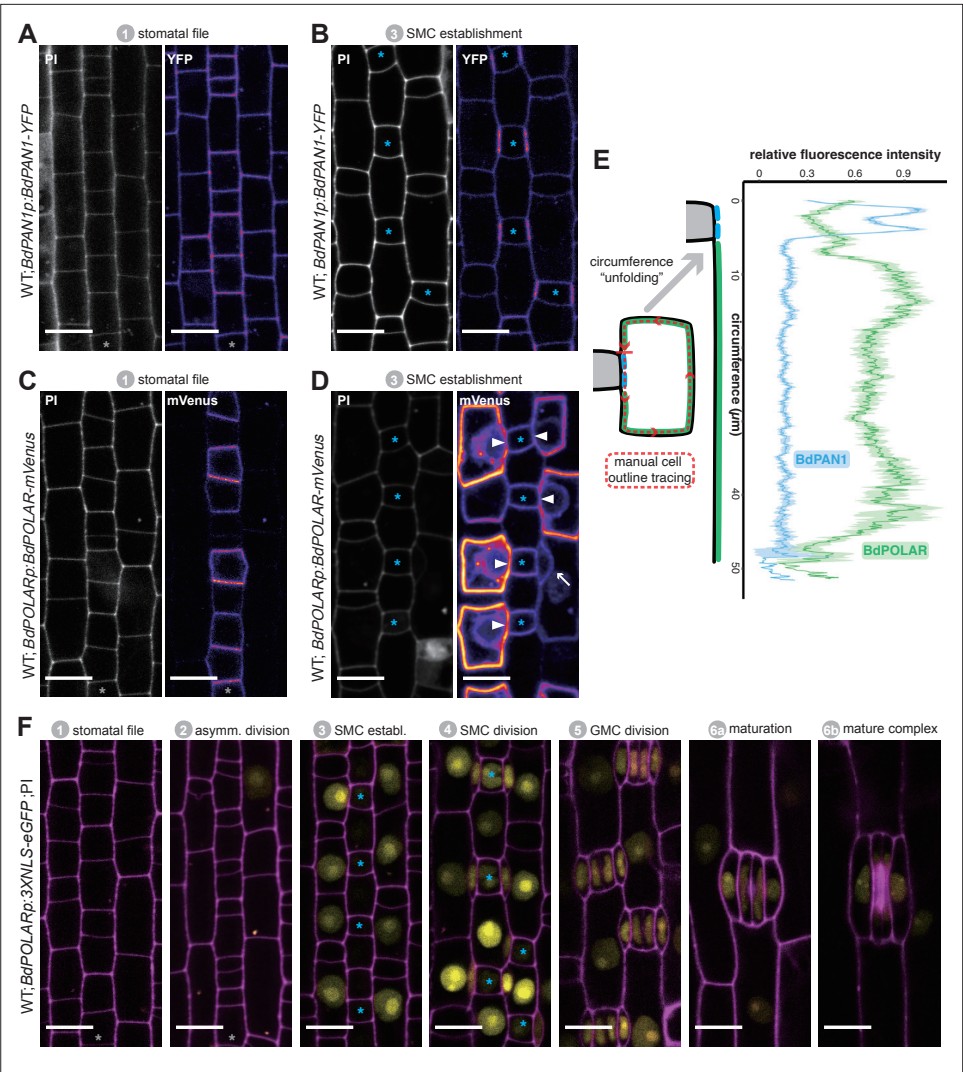

**Figure 2.** BdPAN1 and BdPOLAR display opposite, reciprocal polarisation in subsidiary mother cells (SMCs). (**A, B**) *BdPAN1p:BdPAN1-YFP* reporter expression in stage 1 stomatal files (**A**) and during SMC stage 3 (**B**). Images of propidium iodide (PI)-stained cell outlines (left), and fluorescence intensity of YFP channel (right) (**C, D**). *BdPOLARp:BdPOLAR-mVenus* reporter expression in stage 1 stomatal files (**C**) and during SMC stage 3 (**D**). Images of PI-stained cell outlines (left), and fluorescence intensity of mVenus channel (right). The absence of *BdPOLAR* at the guard mother cell (GMC)/SMC interface is indicated with white arrowheads. The single white arrow indicates a newly formed subsidiary cell; note that BdPOLAR-mVenus dissociated from the SMC distal polarity domain. (**E**) Manually traced, normalised fluorescence intensity at the SMC plasma membrane of BdPAN1-YFP lines (n=13) and BdPOLAR-mVenus lines (n=8). Average fluorescence intensity with standard error is plotted as a function of the 'unfolded' SMC starting with the GMC/SMC interface as indicated by the model. Note that the signal gets noisy towards the end as SMCs are differently sized. Only SMCs with a length/width ratio (LWR) >0.8 for BdPAN1-YFP and LWR >0.9 for BdPOLAR-mVenus were included. (**F**) *BdPOLARp:3xNLS-eGFP* reporter expression during stomatal development. Overlay images show *BdPOLAR* transcriptional signals (in yellow) and PI-stained cell outlines in magenta. The stomatal files are indicated with a grey asterisk. GMCs are indicated with blue asterisks. Developmental stages are indicated. Confocal images shown are single focal planes midway from top to bottom. Scale bar, 10 µm.

The online version of this article includes the following source data and figure supplement(s) for figure 2:

**Source data 1.** Quantification of fluorescence intensity of cell outlines of SMCs expressing BdPOLAR-mVenus or BdPAN1-YFP.

*Figure 2 continued on next page*

*Figure 2 continued*

**Figure supplement 1.** BdPAN1 and BdPOLAR expression throughout stomatal development in *Brachypodium distachyon*.

**Figure supplement 2.** BdPAN1, BdPOLAR, and BdTAN1 reporter expression in representative subsidiary mother cells (SMCs).

accumulate at the GMC/SMC interface (*Figure 2B*; *Figure 2—figure supplement 1A*). Together, BdPAN1's expression and polarisation patterns were highly similar to the localisation of *ZmPAN1* as observed by immunolocalisation or translational reporter lines in *maize* (*Cartwright et al., 2009*; *Sutimantanapi et al., 2014*). Polarising BdPAN1 domains initially extended beyond the GMC/SMC interface (e.g. *Figure 2—figure supplement 1A*, third panel, left side of the second GMC from top) suggesting that like in maize the BdPAN1 polarity domain forms in the SMC rather than in the GMC (*Cartwright et al., 2009*; *Sutimantanapi et al., 2014*). In fact, all thus far identified regulators of SMC polarity (e.g. *ZmPAN1/2*, *ZmROP2/9*, and *ZmBRK1*) polarised much like PAN1 towards the GMC/SMC interface (*Facette et al., 2015*; *Humphries et al., 2011*; *Zhang et al., 2012*).

Next, we generated a *BdPOLARp:BdPOLAR-mVenus* reporter line to analyse if BdPOLAR was also polarly localised in SMCs. Strikingly, signal was detected at the distal, the apical, and the basal cell wall and was excluded from the classical polarity domain at the GMC/SMC interface. Thus, BdPOLAR defines a new, opposing polarised domain (*Figure 2E*; *Figure 2—figure supplement 1B*) in SMCs. In addition, unlike BdPAN1-YFP, which was initially expressed in all protodermal cells, *BdPOLAR* was only expressed in the stomatal lineage (*Figure 2F*, *Figure 2—figure supplement 1B*). Before SMCs were specified, weak BdPOLAR-mVenus signal appeared in the stomatal rows with a bias towards the base of stage 1 stomatal lineage cells and distal to the transverse asymmetric division generating GMCs (*Figure 2C*; *Figure 2—figure supplement 1B*). During stage 3 (SMC establishment, *Figure 1A*), a strong and specific signal emerged in SMCs (*Figure 2D*). Initially, the signal appeared in the apical and basal plasma membrane (PM) of SMCs (*Figure 2—figure supplement 1B*) before also occupying the distal PM (*Figure 2D*). Just after the division, BdPOLAR quickly dissociated from the distal PM (*Figure 2D*, arrow; *Figure 2—figure supplement 1B*). The signal then appeared in the GMCs, young GCs, and young SCs but remained weak (*Figure 2—figure supplement 1B*).

To quantify whether proximal BdPAN1 and distal BdPOLAR indeed form opposite polarity domains, we manually traced the outlines of mature SMCs expressing either BdPAN1-YFP or BdPOLAR-mVenus to quantify mean fluorescence intensity along the SMC circumference (*Figure 2E*, *Figure 2—figure supplement 2*; for details see Methods). Averaged and normalised fluorescence intensity along the SMC circumference (starting with the GMC/SMC interface as depicted in the model) strongly indicate that these two polarity domains are indeed opposite (*Figure 2E*). They are, however, not perfectly reciprocal as there is a region between the BdPAN1 domain and the BdPOLAR domain where no signal was detected from either reporter (*Figure 2E*).

The transcriptional *BdPOLAR* reporter (*BdPOLARp:3XNLS-eGFP*) showed *BdPOLAR* promoter activity first in SMCs, and only later in GMCs, and expression was not detected in young stomatal cell files (*Figure 2F*). This suggests that intronic elements may also contribute to BdPOLAR expression and accumulation.

Together, the analysis of translational SMC polarity reporter lines revealed a novel, distal polarity domain in SMCs that seemed to be reciprocally opposed to the well-established, proximal polarity domain at the GMC/SMC interface.

## *BdMUTE* is required for *BdPOLAR* expression

*BdPOLAR* was 12-fold downregulated in *bdmute* compared to WT (*Supplementary file 1*). To confirm that *BdPOLAR* expression is dependent on *BdMUTE*, *BdPOLARp:BdPOLAR-mVenus* was introduced into *bdmute* through transformation. Because different transgenic lines were compared, we set up laser intensities so that the expression pattern in stage 1 stomatal files was comparable between *bdmute* and WT. In WT, we needed four times less laser to visualise BdPOLAR-mVenus signal in SMCs compared to the signal in stage 1 cell files (2 vs 8%; *Figure 3A*). In *bdmute*, however, we used twice as much laser for stage 3 SMCs as for stage 1 stomatal cell files (20 vs 38%) and still did not detect any signal in cells neighbouring the GMCs (*Figure 3B*). Together with the comparative RNA-seq data,

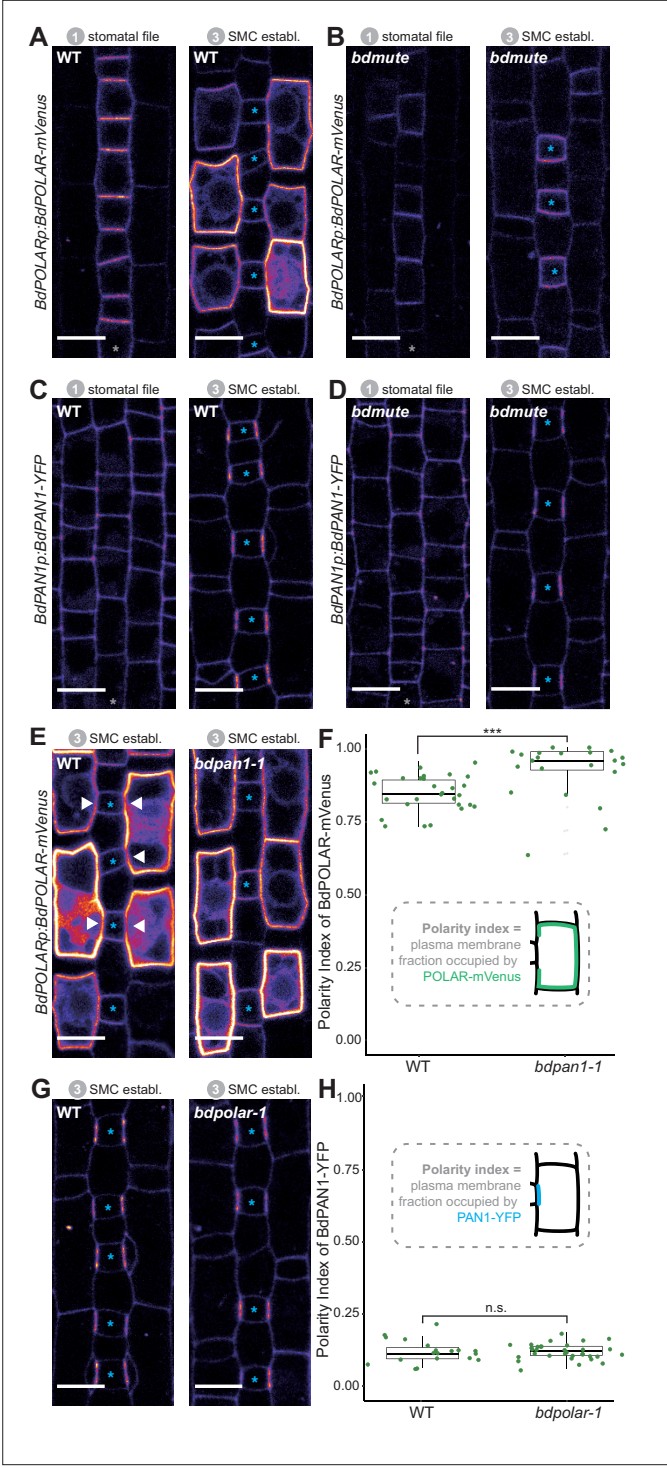

**Figure 3.** BdPOLAR expression requires *BdMUTE,* and BdPOLAR polarisation is controlled by *BdPAN1.* (**A, B**) Fluorescence intensity of *BdPOLARp:BdPOLAR-mVenus* in wild type (WT) (**A**) and in *bdmute* (**B**) at stage 1 and stage 3. (**C, D**) Fluorescence intensity of *BdPAN1p:BdPAN1-YFP* in WT (**C**) and in *bdmute* (**D**) at stage 1 and stage 3. (**E**) Fluorescence intensity of *BdPOLARp:BdPOLAR-mVenus* in WT (left) and *bdpan1-1* (right) at stage 3. The absence of *BdPOLAR-mVenus* at guard mother cell (GMC)/subsidiary mother cell (SMC) interfaces in WT is indicated with white arrowheads. (**F**) Quantification of the polarity index of *BdPOLAR-mVenus* in WT (n=29 SMCs) and *bdpan1-1* (n=22 SMCs). Insets represent that polarity index is the fraction of the plasma membrane (PM) occupied by the polarity protein. (**G**) Fluorescence intensity of *BdPAN1p:BdPAN1-YFP* in WT (left) and *bdpolar-1* (right) at stage 3. (**H**) Quantification of the polarity index of *BdPAN1-YFP* in WT (n=19 SMCs) and *bdpolar-1*

*Figure 3 continued on next page*

*Figure 3 continued*

(n=33 SMCs). Insets represent that polarity index is the fraction of the PM occupied by the polarity protein. Statistical difference was tested with an unpaired Mann-Whitney U-test; \*\*\*=p-value <0.001; n.s.=non-significant. The stomatal files are indicated with grey asterisks. GMCs are indicated with blue asterisks. Confocal images shown are single focal planes midway from top to bottom. Scale bar, 10 μm.

The online version of this article includes the following source data and figure supplement(s) for figure 3:

**Source data 1.** Polarity measurement data.

**Figure supplement 1.** Dosage of BdPOLAR is crucial for its function.

**Figure supplement 1—source data 1.** Quantification of dosage effects of BdPOLAR-mVenus in different genetic backgrounds.

this suggested that *BdMUTE* was indeed required for *BdPOLAR* expression either directly through binding to the *BdPOLAR* promoter or indirectly through the establishment of SMCs and downstream programs.

In contrast, *BdPAN1p:BdPAN1-YFP* was expressed and properly polarised independently of *BdMUTE* (*Figure 3C, D*). This observation suggested that BdPAN1 polarisation was independent of a successful establishment of the SC lineage in lateral cell files and indicated that either biochemical or mechanical signals from the elongating GMC polarise BdPAN1. Alternatively, BdMUTE might also act downstream of BdPAN1 in SMCs.

## *BdPAN1* is required for *BdPOLAR* polarisation but not vice versa

In animal systems, opposing polarity domains can be mutually inhibitory (*Muroyama and Bergmann, 2019*; *Ramalho et al., 2022*; *Nance and Zallen, 2011*), yet such inhibitory relationships have not been identified in the plant kingdom. To test if the opposing BdPAN1 and BdPOLAR polarity domains showed an inhibitory relationship we crossed the BdPOLAR-mVenus and BdPAN1-YFP reporter lines to *bdpan1-1* and *bdpolar-1*, respectively. To quantify the degree of polarisation of BdPOLAR-mVenus and BdPAN1-YFP in WT and mutant SMCs, we employed polarity measurement (POME) to quantify the fraction of the PM occupied by the protein of interest and to determine the polarity index (*Gong et al., 2021a*; *Gong et al., 2021b*). In WT, the broad, distal BdPOLAR polarisation domain showed a much higher average polarity index (i.e. less polarised) than the highly polarised BdPAN1 domain at the GMC/SMC interface (*Figure 3F, H*). In *bdpan1-1*, however, we observed an almost uniform distribution of BdPOLAR-mVenus signal at the periphery of SMCs, where BdPOLAR-mVenus seemed to have invaded the BdPAN1 domain at the GMC/SMC interface (*Figure 3E*). Quantifying the polarity index of BdPOLAR-mVenus in *bdpan1-1* indeed showed a significantly decreased polarisation of BdPOLAR-mVenus (*Figure 3F*). In contrast, BdPAN1-YFP polarised normally at the GMC/SMC interface independent of the presence or absence of functional BdPOLAR (*Figure 3G, H*). Thus, functional BdPAN1 was required for polarised localisation of BdPOLAR, but BdPAN1 localisation was not affected in the *bdpolar-1* mutant background. This suggested that even though BdPAN1 and BdPOLAR show opposing polarisation patterns they exhibit a one-way rather than a mutual inhibition. This might be due to the earlier polarisation of BdPAN1, which accumulated at the GMC/SMC interface before BdPOLAR was expressed in SMCs (*Figure 2—figure supplement 1*).

## Accurate dosage and localisation of BdPOLAR are necessary for functionality

In addition to the non-polarised localisation of BdPOLAR-mVenus in *bdpan1-1*, the expression of BdPOLAR-mVenus strongly enhanced the *bdpan1-1* SC defect frequency (*Figure 3—figure supplement 1A*). 86% of SMCs displayed abnormal formation in *bdpan1-1* plants that expressed BdPOLAR-mVenus compared to only ~30% in *bdpan1-1* plants without BdPOLAR-mVenus expression (*Figure 3—figure supplement 1B*). This suggested that having no BdPOLAR in *bdpan1-1* (i.e. the *bdpolar-1;bdpan1-1* double mutant) or too much and potentially mVenus-stabilised BdPOLAR in *bdpan1-1* had a similar deleterious effect on SC divisions. In fact, even WT plants with strong *BdPOLARp:BdPOLAR-mVenus* expression (~19%) or ubiquitously expressed BdPOLAR-mVenus (~10%) resulted in SC defects further indicating that correct dose and/or stability of BdPOLAR seems to be crucial for its function (*Figure 3—figure supplement 1B, I, J*).

Finally, only weak BdPOLAR-mVenus expression partially rescued *bdpolar-1* defects, while strong BdPOLAR-mVenus expression failed to complement altogether (*Figure 3—figure supplement 1C, D*). It is also conceivable that the lines do complement but create a simultaneous, dosage-dependent overexpression phenotype. To test if the mVenus tag interferes with function, we generated an untagged complementation construct that included the coding sequence, the promoter (–1.1 kb), and the terminator (+1.9 kb) of *BdPOLAR*. Indeed, this construct performed much better at rescuing the *bdpolar-1* phenotype although some remaining defects were still observed in T0 lines (*Figure 3—figure supplement 1K*). A potential caveat of the apparent functional impairment of BdPOLAR when tagged with a fluorescent protein is that the localisation pattern and the temporal dynamics of the tagged reporter line might not fully reflect the spatiotemporal dynamics and/or localisation of endogenous BdPOLAR.

In contrast to BdPOLAR-mVenus, we did not observe a dosage or stability effect of BdPAN1-YFP in WT or *bdpolar-1*, and *BdPAN1p:BdPAN1-YFP* was able to fully complement the *bdpan1-1* mutant phenotype (*Figure 3—figure supplement 1E–H*).

## BdPAN1 promotes nuclear migration in pre-mitotic SMCs to orientate SMC divisions

The phenotypes noticed in *bdpolar-1;bdpan1-1*, and the oppositely polarised domains of BdPOLAR and BdPAN1 suggested that *BdPOLAR* and *BdPAN1* act in parallel pathways and potentially execute different functions during SMC polarisation and division.

It was previously suggested that the ZmPAN1 polarity domain guides pre-mitotic nuclear migration (*Cartwright et al., 2009*; *Facette et al., 2015*; *Zhang et al., 2012*). To test if *BdPAN1* is also required for pre-mitotic nuclear migration in *B. distachyo*n and to determine to what extent *BdPOLAR* contributes to this process, we quantified the distance of the SMC nucleus to the GMC/SMC interface as a proxy for pre-mitotic SMC polarisation and nuclear migration (*Figure 4A*). Images of developing WT leaf zones revealed that indeed most SMCs had polarised nuclei (*Figure 4B*). In *bdpolar-1,* only a few SMCs displayed unpolarised nuclei (*Figure 4B*), but in *bdpan1-1* and *bdpolar-1;bdpan1-1,* many SMCs showed unpolarised nuclei (*Figure 4B*).

To accurately quantify the nuclear position as a function of the SMC's developmental stage, we not only measured the distance (d) from the nuclear centre to the middle of the GMC/SMC interface but also the length-to-width ratio (LWR) of the neighbouring GMC (*Figure 4A*). SMC nuclei were located more proximal to the GMC/SMC interface next to longer and, thus, more mature GMCs (i.e. higher LWR of GMCs; *Figure 4—figure supplement 1A*). We chose a GMC LWR of 0.9 as cut-off since >93% of the SMC nuclei were within 4 µm of the GMC/SMC domain and thus considered polarised in WT (*Figure 4—figure supplement 1C*). In WT, the average distance of SMC nuclei to GMCs with an LWR >0.9 was 3.19 µm— the lowest distance value of all genotypes. The average distance of SMC nuclei was significantly increased in *bdpolar-1* (3.48 µm; *Figure 4C*, *Figure 4—figure supplement 1C*) but almost 80% of the nuclei were still within 4 µm of the GMC/SMC interface. In *bdpan1-1,* the average distance was clearly increased (3.78 µm; *Figure 4—figure supplement 1C*) and more than 35% of the SMC nuclei were beyond the 4 µm cut-off (*Figure 4C*) suggesting that *BdPAN1* indeed affected nuclear migration more than *BdPOLAR*. Finally, the double mutant *bdpolar-1;bdpan1-1* displayed an almost identical nuclear migration defect (3.89 µm, and >37% unpolarised nuclei; *Figure 4—figure supplement 1C*) as *bdpan1-1* single mutants (*Figure 4C*). This suggested that nuclear migration defects in *bdpolar-1;bdpan1-1* were mostly caused by the loss-of-function of *BdPAN1,* and indicated that nuclear migration is promoted by BdPAN1 rather than BdPOLAR activity (*Figure 4C*). In addition, the fraction of unpolarised SMC nuclei in *bdpan1-1* and the *bdpolar-1;bdpan1-1* mutants are similar, yet the defective SMC divisions are twice as frequent in the double mutant. This suggested that the exaggerated ACD defects in the double mutant might not solely be induced by deficient nuclear migration.

## BdPOLAR is involved in cortical division site orientation

As defective nuclear migration cannot explain the severe SC division defects in *bdpolar-1;bdpan1-1*, another, yet unknown pathway must also be affected and likely involves *BdPOLAR*. We not only found an absence of BdPOLAR-mVenus signal at the GMC/SMC interface but also at the cortical division sites just above and below the GMC/SMC interface (*Figure 2D*). We, therefore, hypothesised that

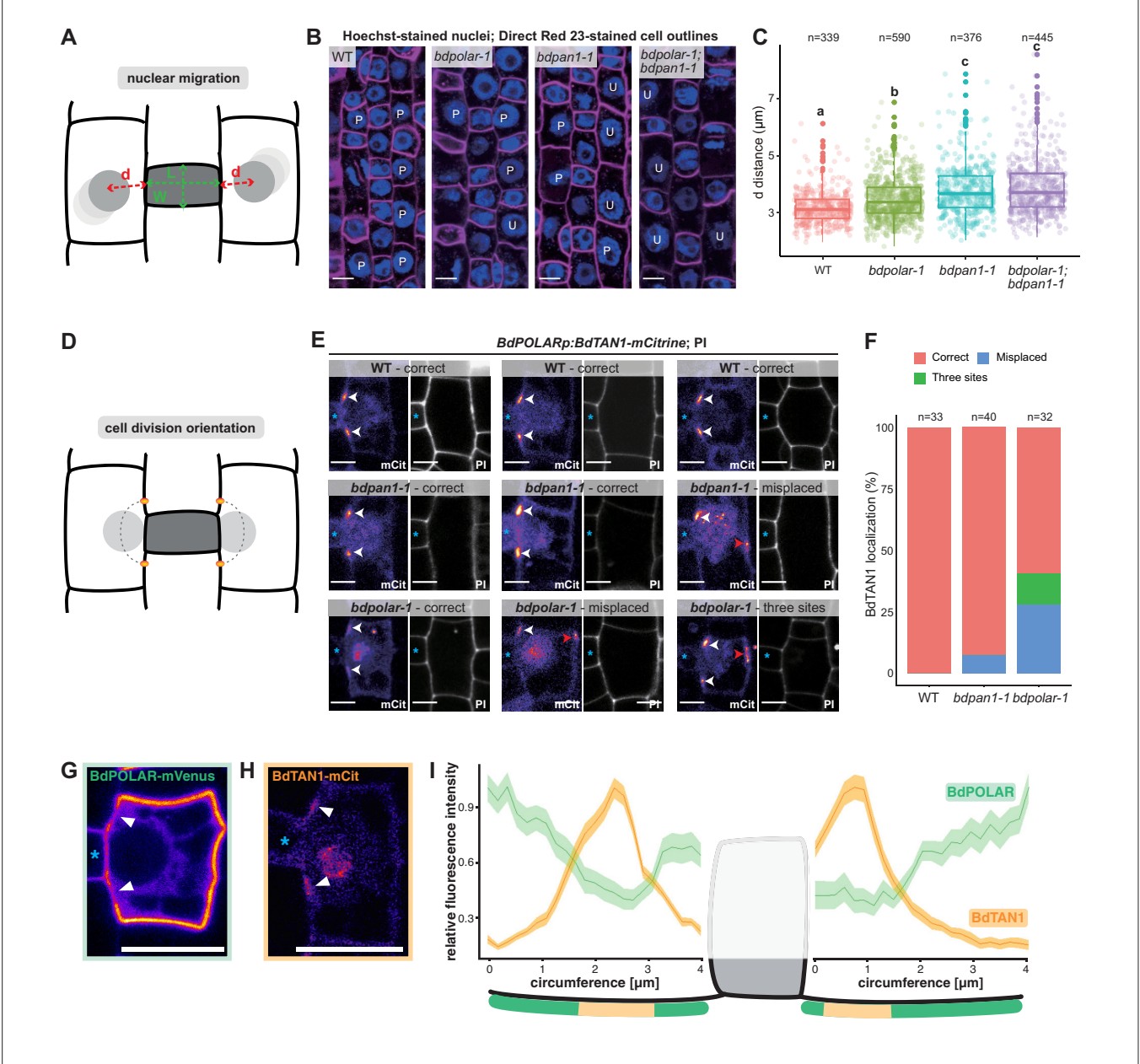

**Figure 4.** BdPAN1 promotes nuclear migration, and BdPOLAR guides cortical cell division orientation in subsidiary mother cells (SMCs). (**A**) Schematic of the quantification of nuclear migration. L: guard mother cell (GMC) length; W: GMC width; d: the distance between the nuclear centre and the middle of the GMC/SMC interface. (**B**) Single confocal plane images of Hoechst-stained nuclei and Direct Red 23-stained cell outlines of stage 3 to stage 4 SMCs in wild type (WT), *bdpolar-1*, *bdpan1-1*, and *bdpolar-1;bdpan1-1*; P, polarised nucleus; U, unpolarised nucleus. Scale bar, 10 µm. (**C**) Quantification of d in SMCs of WT, *bdpolar-1*, *bdpan1-1*, and *bdpolar-1;bdpan1-1*; only SMCs flanking GMCs with a length/width ratio >0.9 are shown. Numbers of SMCs measured are indicated; n=20–28 individuals per genotype. Samples were compared using a one-way ANOVA and post hoc Tukey test for multiple comparisons; different letters indicate significant differences (p<0.05). (**D**) Schematic showing future division plane and expected BdTAN1 localisation (orange) at the cortical division site. (**E**) *BdPOLARp:BdTAN1-mCitrine* expressed in WT, *bdpan1-1,* and *bdpolar-1*; three different SMCs are shown per genotype. Correct BdTAN1-mCitrine localisation is indicated with white arrowheads; misplaced cortical division sites are indicated with red arrowheads. Note that BdTAN1-mCitrine signal is also present in SMC nuclei and nucleoli. GMCs are indicated with blue asterisks. Scale bar, 5 µm. (**F**) Percentages of correct, misplaced, and three cortical division sites in WT, *bdpolar-1,* and *bdpan1-1*; the numbers of SMCs analysed are indicated; n=8–10 individuals for WT and *bdpan1-1* and 5 individuals for *bdpolar-1*. (**G, H**) Representative image of an SMC expressing BdPOLAR-mVenus (**G**) or BdTAN1-mCitrine (**H**). Arrowheads point to the regions where BdPOLAR-mVenus signal is absent (**H**), and BdTAN1-mCitrine signal is present (**G**). Scale bar, 10 µm. (**I**) Manually traced, normalised fluorescence intensity at the SMC plasma membrane of BdPOLAR-mVenus lines (n=8, same data as in *Figure 2E*) and BdTAN1-mCitrine lines (n=13). Average fluorescence intensity normalised to the average max grey intensity, 4 µm before and after

*Figure 4 continued on next page*

*Figure 4 continued*

the GMC were plotted; fluorescence intensity values along the GMC/SMC interface were removed as GMC length differs for each measurement. The schematic model below the plot indicates what is shown. Confocal images shown are single focal planes midway from top to bottom.

The online version of this article includes the following source data and figure supplement(s) for figure 4:

**Source data 1.** Quantification of cellular defects in *bdpolar-1*, *bdpan1-1,* and *bdpolar-1;bdpan1-1*.

**Figure supplement 1.** *BdPAN1* controls nuclear polarisation.

**Figure supplement 1—source data 1.** Quantification of cellular parameters and cellular defects in *bdpolar-1*, *bdpan1-1,* and *bdpolar-1;bdpan1-1*.

**Figure supplement 2.** Cell division capacity is increased in subsidiary mother cell (SMC) polarity mutants.

**Figure supplement 2—source data 1.** Quantification of division capacity in *bdpolar-1* and *bdpan1-1*.

---

*BdPOLAR* might have a role in specifying cortical division sites and cell division orientation. To test this possibility, we generated *BdTANGLED1 (BdTAN1)-mCitrine* reporter lines specifically expressed in dividing SMCs and GMCs (using the *BdPOLAR* promoter), which stably marked cortical division sites throughout mitosis (**Figure 4E**) much like the maize homologue (**Martinez et al., 2017**). In *B. distachyon*, however, BdTAN1-mCitrine also shows weak expression in the nucleus and nucleolus (**Figure 4E**), which we could use to quantify nuclear migration (**Figure 4H**, **Figure 4—figure supplement 1F**). Manual tracing and comparison of BdPOLAR-mVenus and BdTAN1-mCitrine occupancy at the SMC circumference confirmed that indeed the BdPOLAR-mVenus signal is not only absent from the BdPAN1 domain at the SMC/GMC interface (**Figure 2D**) but also at the future division site where BdTAN1 localises to (**Figure 4G–I**). In WT SMCs, BdTAN1 always localised above and below the GMC/SMC interface, correctly predicting the future SC division plane (100%, n=33; **Figure 4E, F**). In *bdpan1-1* SMCs, BdTAN1 is still mostly localised to the 'wild-type'-like proximal division site independent of nuclear polarisation (92.5%, n=40; **Figure 4E, F**). In *bdpolar-1* SMCs, however, we found that in 28.1% of cases (n=32) BdTAN1 is mis-localised to the proximal and distal PM predicting a transverse, oblique rather than longitudinal, curved division plane. In addition, in 12.5% of *bdpolar-1* SMCs, BdTAN1 marked three rather than just two cortical sites (**Figure 4E, F**). This indeed suggested a role for *BdPOLAR* in orienting cell division planes in SMCs.

To exclude that our observation regarding the diverse cellular roles of BdPAN1 and BdPOLAR stems from growth artefacts and/or individual outliers, we repeated the cellular phenotyping on a set of simultaneously grown plants. We grew WT, *bdpolar-1,* and *bdpan1-1* plants, each expressing *BdPOLARp:BdTAN1-mCitrine*, on common nutrient agar plates. We then quantified the distance between SMC nucleus and GMC/SMC interface and cortical TAN1-mCitrine-based division site specification in the same SMCs of the second leaf (**Figure 4—figure supplement 1E, F**). We then used the third leaf at 19 dag of these exact individualsto quantify SC division defects of the mature leaf (for details see Methods). Using this approach, we could confirm the nuclear migration defects in *bdpan1-1* (**Figure 4C**, **Figure 4—figure supplement 1F**). Yet unlike before (**Figure 4C**), we did not find any nuclear migration defects in *bdpolar-1* compared to WT strongly suggesting that there are no or very weak nuclear migration defects in *bdpolar-1* (**Figure 4—figure supplement 1F**). We could also confirm a more severe defect in cortical division site localisation in *bdpolar-1* (16.5%) compared to *bdpan1-1* (6.5%; **Figure 4—figure supplement 1E**). Finally, we found that both mutant lines in this set exhibited a similar frequency of SC division defects (37% in *bdpolar-1* and 44% in *bdpan1-1*; **Figure 4—figure supplement 1D**). This indicates that indeed BdPAN1 promotes nuclear migration and that cortical division site defects are exaggerated in *bdpolar-1*. It remains unclear, however, why the comparatively low frequency of wrongly specified cortical division sites leads to a relatively high frequency of abnormally formed SCs (see Discussion). Finally, we investigated if *BdPOLAR* has a role in regulating cell division potential like its homologue in *Arabidopsis* (**Figure 4—figure supplement 2A–C**). We found that individual SMCs in both *bdpolar-1* and *bdpan1-1* can undergo more than one round of misoriented ACDs (**Figure 4—figure supplement 2B, C**). Quantifying rounds of cell divisions per individual SMC in both mutant backgrounds revealed that in *bdpolar-1* almost 75% of the SMCs divided more than one time, whereas only 55% of the SMCs in *bdpan1-1* divided more than once (**Figure 4—figure supplement 2B, C**). In *bdpolar-1*, we even observed a small fraction of SMCs (1.5%) that exhibited four rounds of aberrant cell division, which we never saw in *bdpan1-1* (**Figure 4—figure supplement 2B, C**). On average, SMCs divided 1.9 times in *bdpolar-1* but only 1.6 times in *bdpan1-1*.

Successful recruitment of WT-like SCs seemed to occur at later stages in *bdpolar-1* compared to *bdpan1-1*. We thus quantified the LWR of GMCs that successfully recruited one or two WT-like SCs as a proxy for developmental stage/maturity in WT, *bdpan1-1*, *bdpolar-1,* and *bdpolar-1;bdpan1-1* (*Figure 4—figure supplement 2D, F*). We found that the LWR of GMCs that recruited one or two SCs was always higher in *bdpolar-1* and *bdpolar-1;bdpan1-1* compared to *bdpan1-1* andWT (*Figure 4— figure supplement 2E, G*). This suggested that GMCs that successfully recruited SCs were developmentally more mature in *bdpolar-1* and *bdpolar1-1* compared to *bdpan1-1* and WT. This could be due to a potential developmental delay in *bdpolar-1*, or indicate that more rounds of divisions occurred until successful SC recruitment was completed. Together, these findings suggested that *BdPOLAR* might affect cell division potential. Yet, repeated rounds of cell division in SMCs were also observed in *bdpan1-1* suggesting that repetitive divisions might be an inherent consequence of misoriented SMC divisions rather than a gene-specific effect.

## Morphologically aberrant SCs negatively impact stomatal gas exchange

We have previously shown that *bdmute* stomatal complexes that failed to specify and recruit SCs altogether displayed severely impaired stomatal dynamics and gas exchange capacity (*Raissig et al., 2017*). In *bdpolar-1;bdpan1-1* double mutants also ~82% of SCs fail to form correctly (*Figure 1D*). Yet in *bdpolar-1;bdpan1-1*, the neighbouring cells presumably adopted SC lineage identity through the action of *BdMUTE* but failed to properly divide and form SCs. To test stomatal opening and closing speediness, we performed gas exchange measurements in a changing light regime. We found that both single mutants and the double mutant *bdpolar-1;bdpan1-1* showed lower absolute stomatal conductance ($g_{sw}$; *Figure 5A, C and E*) that seemed proportional to the number of aberrantly formed SCs. This was not linked to reduced stomatal density (*Figure 5—figure supplement 1A*) but rather stomatal size (*Figure 5—figure supplement 1B*), as GCs without SCs tend to be shorter and accordingly have shorter pores (*Nunes et al., 2022*). In contrast, the analysis of relative $g_{sw}$ (values normalised to maximum stomatal conductance per genotype and individual), which visualises differences in stomatal opening and closing kinetics, showed slower stomatal movements in the double mutant only (*Figure 5B, D and F*; *Figure 5—figure supplement 1C, D, E*). This indicated that a majority of SCs must be defective to affect speed. In addition, the effect on speediness seemed less severe in *bdpolar-1;bdpan1-1* compared to *bdmute* (*Raissig et al., 2017*) suggesting that the mis-divided cells flanking the GCs in *bdpolar-1;bdpan1-1* might retain residual SC functionality.

## Discussion

In this study, we identified *BdPOLAR* as a new, stomatal lineage-specific polarity factor that is required for the formative cell division that yields the functionally high relevant stomatal SCs in the model grass *B. distachyon*. Strikingly, *BdPOLAR* defined a new, distal polarity domain that was reciprocal to the proximal BdPAN1 polarity domain at the GMC/SMC interface (*Figure 6A*). The BdPOLAR domain seemed functionally distinct from the BdPAN1 domain, yet BdPOLAR's polarised localisation was tightly linked to BdPAN1 (*Figure 6B*).

In *Arabidopsis*, POLAR, like the BRX family and BASL, polarises distal to the nascent division plane, with BASL acting as a landmark for pre-mitotic nuclear migration away from the domain (*Muroyama et al., 2020*). In grasses, however, the positional cue for nuclear migration seems to be linked to the PAN1 polarity domain at the GMC/SMC interface, which is proximal to the division plane and is required to attract rather than repel the migrating SMC nucleus pre-mitotically (*Figure 6B*; *Cartwright et al., 2009*; *Facette et al., 2015*; *Humphries et al., 2011*). In *B. distachyon*, the PAN1 polarity domain is formed independently of SMC identity (i.e. also in *bdmute*) and, thus, seems to be primarily following mechanical or biochemical signals from the rapidly elongating GMCs (*Giannoutsou et al., 2016*; *Livanos et al., 2016*; *Livanos et al., 2015*; *Nunes et al., 2020*). Furthermore, the PAN1 polarity domain is established significantly before the SMC division and persists significantly after the SMC division until late stages. This suggests that the PAN1 domain is a 'primary' polarity domain that reads SC lineage independent polarity cues and is stably positioned at the GMC/SMC interface throughout stomatal development. Finally, BdPAN1 is initially expressed in all protodermal cells, and only later executes an SC-specific, polarised accumulation.

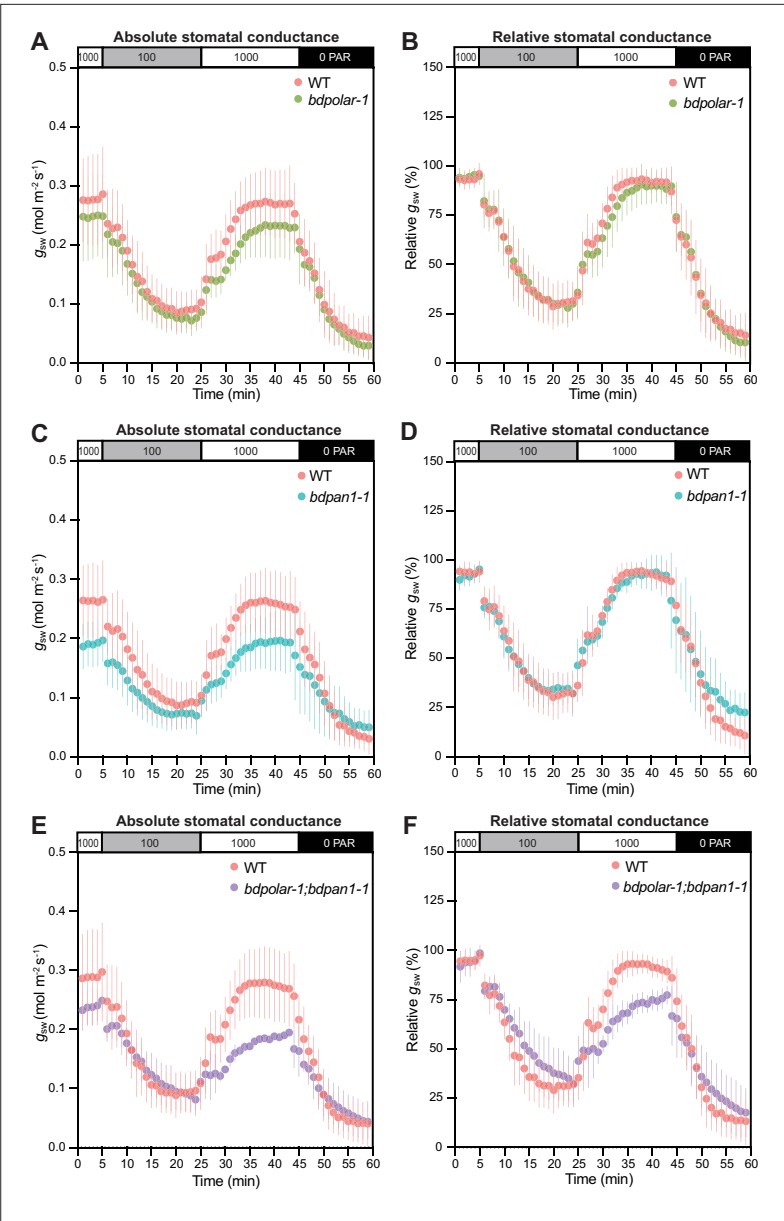

**Figure 5.** Wrongly divided subsidiary cells affect stomatal gas exchange levels and kinetics. (**A, C, and E**) Absolute stomatal conductance ($g_{sw}$) in response to light transitions (1000 photosynthetic active radiation [PAR] - 100 PAR - 1000 PAR - 0 PAR) in *bdpolar-1* (**A**), *bdpan1-1* (**C**), and *bdpolar-1; bdpan1-1* (**E**) compared to paired wild-type (WT) measurements. (**B, D, and F**) Relative stomatal conductance ($g_{sw}$ normalised to highest $g_{sw}$ observed) in response to light transitions (1000 PAR - 100 PAR - 1000 PAR - 0 PAR) in *bdpolar-1* (**B**), *bdpan1-1* (**D**), and *bdpolar-1; bdpan1-1* (**F**) compared to paired WT measurements. n=5 individuals per mutant genotype and 7 WT individuals. Error bars represent standard deviation.

The online version of this article includes the following source data and figure supplement(s) for figure 5:

**Source data 1.** Stomatal conductance data in wild type, *bdpolar-1, bdpan1-1,* and *bdpolar-1;bdpan1-1*.

**Figure supplement 1.** The stomatal density and guard cell (GC) length in wild type (WT), *bdpolar-1, bdpan1-1,* and *bdpolar-1;bdpan1-1*.

**Figure supplement 1—source data 1.** Stomatal anatomical parameters in wild type, *bdpolar-1, bdpan1-1,* and *bdpolar-1;bdpan1-1*.

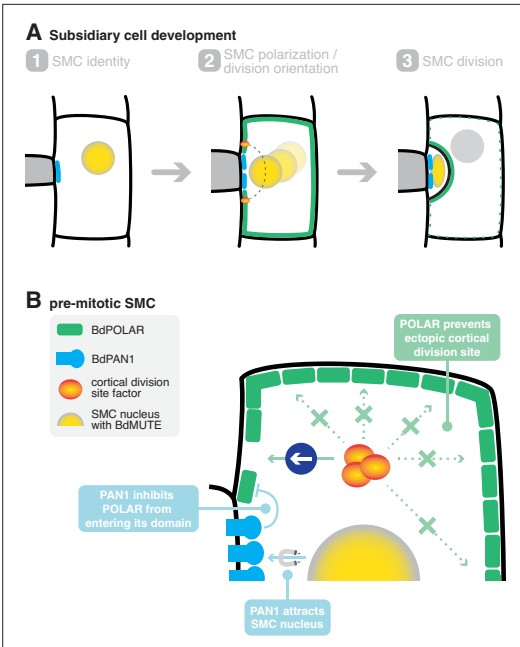

**Figure 6.** A mechanistic model of subsidiary mother cell (SMC) polarisation by the two oppositely polarised BdPOLAR and BdPAN1 domains. (**A**) Stages of subsidiary cell (SC) development. (1) SMC identity is established by *BdMUTE* and BdPAN1 polarises at the guard mother cell (GMC)/SMC interface independent of SMC establishment. (2) BdPOLAR is strongly expressed in SMCs and localises at the apical, basal, and distal plasma membrane (PM) of SMCs and is excluded from the BdPAN1 domain. The SMC nucleus migrates towards the GMC/SMC interface, and the cortical division sites are established above and below the GMC/SMC interface. (3) An asymmetric cell division generates a smaller SC and a larger pavement cell. BdPOLAR quickly dissociates from the PM and reorients towards the newly formed PM. (**B**) Cellular mechanisms of BdPAN1 and BdPOLAR in pre-mitotic SMCs. BdPAN1 guides the migration of the SMC nucleus towards its own domain. BdPAN1 actively inhibits BdPOLAR from entering its domain, either biochemically or by steric hindrance. Distal BdPOLAR prevents cortical division site factors from ectopically binding and forces it to form specifically where BdPOLAR protein level is low.

*BdPOLAR*, on the other hand, is specifically expressed in the stomatal lineage. Its expression is strongest in SMCs, which depends on *BdMUTE* establishing SC identity. This suggests that unlike the PAN1 domain, which seems to respond to intercellular polarisation cues, BdPOLAR might be a cell-autonomous polarity factor linked to a specific cell fate. In addition, BdPOLAR is the first SMC polarity factor that forms a polarity domain distal to the future division plane in SMCs. BdPOLAR shows strong but transient polarised localisation before it is quickly dissociated from the PM after completion of the ACD (*Figure 6A and B*). The transient nature of BdPOLAR localisation might underlie its sensitivity to dosage and/or altered protein stability potentially explaining the incomplete rescue of the *bdpolar-1* phenotype by the translational reporter. Similarly, YFP-ZmROP2 reporter lines in maize showed SCs defects likely due to dosage and mislocalisation effects of YFP-ZmROP2 in SMCs (*Humphries et al., 2011*). Much like for BdPOLAR-mVenus, YFP-ZmROP2 expression levels and SC division defects were correlated (*Humphries et al., 2011*).

Strikingly, the distally polarised localisation of BdPOLAR depends on BdPAN1 (*Figure 6B*) and in the absence of a functional BdPAN1 domain, BdPOLAR was able to invade the GMC/SMC interface. This suggests that the BdPAN1 domain either sterically hinders BdPOLAR from entering the GMC/SMC interface by prior occupation of this domain or it induces localised degradation of BdPOLAR through protein modification. This supports a hierarchical model, where a cell-fate-independent polarity domain (BdPAN1) polarises a cell-fate-dependent polarity domain (BdPOLAR). Finally, much like the PAN homologues, BdPOLAR then persists in young GCs and SCs. While ZmPAN2 was shown to have a role in SC morphogenesis (*Sutimantanapi et al., 2014*), there is no obvious role for the grass PAN1 homologues or BdPOLAR in the SC lineage after the SC division. A role for BdPAN1 and BdPOLAR in the GC lineage is difficult to assess because absence of SCs affected GC length and morphology (*Figure 5—figure supplement 1B*; *Raissig et al., 2017*).

In *B. distachyon*, *BdPOLAR* was initially expressed in the GC lineage and seemed to polarise to the base of dividing cells likely following similar polarity cues as *AtBASL* when it was ectopically expressed in *B. distachyon* (*Nir et al., 2022*). Functionally, however, BdPOLAR played no role in this first ACD that generated the GMC. Upon *BdMUTE*-dependent specification of SMCs, BdPOLAR is redeployed from the GC to the SC lineage, where it is functionally required during ACD. The absence of BdPOLAR signal at cortical division sites and cortical division site mislocalisation revealed by *BdTAN1* reporter imaging in *bdpolar-1* suggested a role in placing and orienting the cortical division site (*Figure 6B*). The rather uniquely positioned cortical division sites in SMCs and strikingly curved SC wall (*Figure 6A*)

might underlie the requirement for not only one but two polarity domains, one of which is cell-fate-specific and orients this peculiar division. The mechanism by which *BdPOLAR* affects division plane orientation is yet to be determined. We propose that the BdPOLAR domain actively repels microtubules or cortical division site factors like DCD1 and ADD1 (*Wright et al., 2009*) and/or phragmoplast orienting kinases (*Herrmann et al., 2018*; *Lipka et al., 2014*; *Müller et al., 2006*; *Figure 6B*). Temporally, however, it seems to act before cell plate formation and phragmoplast expansion and guidance, which are controlled by OPAQUE1/DISCORDIA2 in SMCs (*Nan et al., 2021*).

A relevant open question is why we observed a much lower frequency of mislocalised cortical division sites compared to the relatively high frequency of aberrant SCs particularly in *bdpan1*. In *bdpan1*, we observed <10% cortical division site defects but ~40% defective SCs. In particular, defective nuclear migration in *bdpan1-1* still seemed to mostly allow normal formation of cortical division sites. Conversely in maize, nuclear migration defects seem to be strongly linked to misspecified PPBs (*Arif Ashraf et al., 2022*). This could indicate that defects in the proximal polarity domain in *B. distachyon* also affected later cellular events like phragmoplast formation or guidance. Alternatively, it is conceivable that in *B. distachyon* the extending phragmoplast 'short-cuts' more readily to the distal wall when polarity is compromised. SMC geometry in maize and *B. distachyon* is quite different, which can influence how plant cells divide (e.g. *Besson and Dumais, 2011*; *Martinez et al., 2018*; *Müller, 2019*). In maize, SMCs tend to be much wider relative to their length, whereas *B. distachyon* SMCs are much narrower, which could favour phragmoplast 'short-cuts' when nuclei are not properly migrated to the GMC/SMC interface. In addition, the maize SMC walls just above and below the GMC/SMC interface seem often to be at an ~45° angle with respect to the GMC/SMC wall. This might facilitate proper phragmoplast extension even in polarity-compromised SMCs in maize compared to *B. distachyon*, where the SMC walls above and below GMC/SMC interface are hardly at an angle at all.

In addition to *BdPOLAR*'s role in cell division orientation, it might also affect cell division capacity. In *Arabidopsis*, AtPOLAR interacts and sequesters the GSK3-like kinase BIN2; this results in disinhibition of the stomatal lineage initiating and ACD regulating transcription factor SPEECHLESS (SPCH) before mitosis (*Houbaert et al., 2018*; *MacAlister et al., 2007*). Right at the onset of mitosis, a BSL family phosphatase is recruited to the polarity crescent in meristemoids, which releases BIN2 (*Guo et al., 2021a*). This enables BIN2 to enter the nucleus of the larger daughter cell after the ACD, where it can downregulate SPCH and inhibit ACD potential (*Guo et al., 2021a*; *Houbaert et al., 2018*). Accordingly, mutants affecting several POLAR family members in *Arabidopsis* show reduced cell division capacity in meristemoids. In grasses, a mutation in *BdPOLAR* alone was sufficient to reveal the family's functional relevance, yet *bdpolar-1* SMCs exhibited an increased rather than a decreased cell division capacity. Whether this reversed functionality also involves the sequestration of signalling components that directly or indirectly affects the cell cycle or is just a consequence of misoriented division planes remains elusive. It is possible, for example, that properly migrated SMC nuclei at the GMC/SMC interface in *bdpolar-1* but not in *bdpan1-1* might positively affect cell division capacity. Importantly, excessive cell division in wrongly dividing SMCs was also observed in *bdpan1-1* albeit at a lower frequency. This indicates that division capacity is affected in wrongly dividing SMCs independent of the mutated genetic pathway. How normally formed SCs inhibit additional divisions remains elusive.

SCs functionally contribute to the fast opening and closing kinetics of grass stomata by acting as water and ion reservoirs and by providing mechanical support to GC movement through inverse turgor regulation (*Franks and Farquhar, 2007*; *Gray et al., 2020*; *Nunes et al., 2020*; *Spiegelhalder and Raissig, 2021*). Compared to *bdmute* plants, where the SC lineage is never established, *bdpolar-1;bdpan1-1* showed an overall milder defect in respect to stomatal opening and closing speediness (*Raissig et al., 2017*). This could be due to fewer SCs being defective in *bdpolar-1;bdpan1-1* (82%) compared to *bdmute* (almost 100%), which also results in a reduced effect on GC size and shape. Also, *bdmute* shows lower stomatal density (~30% of stomata abort), which is correlated with slower stomatal kinetics in *B. distachyon* (*Nunes et al., 2022*). Alternatively, and unlike in *bdmute*, wrongly divided SCs in *bdpolar-1;bdpan1-1* might have acquired and retained residual SC identity thus expressing appropriate ion channels, signalling modules or cell wall modifiers, which could positively influence stomatal kinetics. If partial SC-like functionality did not require anatomical changes to the stomatal complex, then stomatal speediness could be more easily engineered in crops that form stomata without SCs.

In conclusion, we identified a novel, cell-fate-dependent, distal polarity domain that is polarised by a SC lineage-independent, proximal polarity domain. Apparently, the peculiar, highly asymmetric division of SMCs seems to require two connected, yet functionally diverse polarity domains to faithfully form this highly relevant cell type for efficient gas exchange in grasses.

# Materials and methods

**Key resources table**

| Reagent type (species) or resource | Designation | Source or reference | Identifiers | Additional information |
|---|---|---|---|---|
| Strain, strain background (*Brachypodium distachyon*) | *B. distachyon* Bd21-3 v1 | Phytozome | Phytozome genome ID: 378 | Genetic background for all transgenic and mutant lines |
| Strain, strain background (*Agrobacterium tumefaciens*) | AGL1 | Provided by John Vogel, JGI-DOE; *Bragg et al., 2015* | AGL1 | Chemically competent cells |
| Gene (*B. distachyon*) | *BdPOLAR* | Phytozome | BdiBd21-3.3G0715200 | |
| Gene (*B. distachyon*) | *BdPAN1* | Phytozome | BdiBd21-3.3G0526300 | |
| Genetic reagent (*B. distachyon*) | *bdmute-1* | Raissig et al. 2007 | | EMS-mutagenised |
| Genetic reagent (*B. distachyon*) | *bdpan1-1* | This paper | | EMS-mutagenised, details see Methods below, available upon request from our lab |
| Genetic reagent (*B. distachyon*) | *bdpolar-1; bdpolar-2; bdpolar-3* | This paper | | CRISPR/Cas9, details see Methods below, available upon request from our lab |
| Genetic reagent (*B. distachyon*) | *BdPOLARp:BdPOLAR-mVenus* in wild type/*bdmute* | This paper | | Tissue culture, details see Methods below, available upon request from our lab |
| Genetic reagent (*B. distachyon*) | *BdPOLARp:BdPOLAR-mVenus* in *bdpolar-1/bdpan1-1* | This paper | | Crossing, details see Methods below, available upon request from our lab |
| Genetic reagent (*B. distachyon*) | *BdPOLARp:3XNLS-eGFP/ZmUbiIp:BdPOLAR-mVenus/BdPAN1p:BdPAN1-YFP* in wild type | This paper | | Tissue culture, details see Methods below, available upon request from our lab |
| Genetic reagent (*B. distachyon*) | *BdPAN1p:BdPAN1-YFP* in *bdmute* | This paper | | Tissue culture, details see Methods below, available upon request from our lab |
| Genetic reagent (*B. distachyon*) | *BdPAN1p:BdPAN1-YFP* in *bdpan1-1/bdpolar-1* | This paper | | Crossing, details see Methods below, available upon request from our lab |
| Genetic reagent (*B. distachyon*) | *BdPOLARp:BdTAN1-mCitrine* in wild type/*bdpan1-1/bdpolar-1* | This paper | | Tissue culture, details see Methods below, available upon request from our lab |
| Recombinant DNA reagent | *BdPOLARp:BdPOLAR-mVenus* (plasmid) | This paper | | Greengate construct in pGGZ004, details see Methods below, available upon request from our lab |
| Recombinant DNA reagent | *BdPAN1p:BdPAN1-YFP* (plasmid) | This paper | | Greengate construct in pGGZ004, details see Methods below, available upon request from our lab |
| Recombinant DNA reagent | *BdPOLARp:3XNLS-eGFP* (plasmid) | This paper | | Greengate construct in pGGZ004, details see Methods below, available upon request from our lab |
| Recombinant DNA reagent | *ZmUbiIp:BdPOLAR-mVenus* (plasmid) | This paper | | Greengate construct in pGGZ004, details see Methods below, available upon request from our lab |

*Continued on next page*

*Continued*

| Reagent type (species) or resource | Designation | Source or reference | Identifiers | Additional information |
|---|---|---|---|---|
| Recombinant DNA reagent | *BdPOLARp:BdTAN1-mCitrine* (plasmid) | This paper | | Gibson construct in pIPKb001t Greengate construct in pGGZ004, details see Methods below, available upon request from our lab |
| Commercial assay, kit | RNeasy Plant Mini kit | Qiagen | | |
| Commercial assay, kit | LR recombination | Invitrogen | | |
| Commercial assay, kit | DNeasy Plant Mini Kit | Qiagen | Cat# 69104 | |
| Commercial assay, kit | Mini-prep Kit NucleoSpin Plasmid Kit | Macherey-Nagel | Cat# 740588.250 | |
| Commercial assay, kit | NucleoSpin Gel and PCR Clean-up Kit | Macherey-Nagel | Cat#740609.250 | |
| Commercial assay, kit | Gibson Assembly Master Mix Kit | NEB | Cat# M5510A | |
| Chemical compound, drug | Q5 High-Fidelity DNA Polymerase | NEB | Cat# M0491L | |
| Chemical compound, drug | Taq DNA Polymerase | NEB | Cat# M0273X | |
| Chemical compound, drug | T4 DNA ligase | Thermo Fisher Scientific | Cat# EL0014 | |
| Chemical compound, drug | FastDigest Eco31I | Thermo Fisher Scientific | Cat# FD0294 | |
| Chemical compound, drug | HindIII | Thermo Fisher Scientific | Cat# ER0501 | |
| Chemical compound, drug | OliI | Thermo Fisher Scientific | Cat# ER1631 | |
| Chemical compound, drug | Hoechst 33342 | Thermo Fisher Scientific | Cat# 62249 | |
| Chemical compound, drug | Direct red 23 | Sigma-Aldrich | Cat# 3441-14-3 | |
| Chemical compound, drug | Propidium iodide (PI) | Thermo Fisher Scientific | | 10 µg/ml |
| Software, algorithm | R v.3.6.0 | *R Development Core Team, 2022* | https://cran.r-project.org/bin/ windows/base/old/3.6.0/ | |
| Software, algorithm | CRISPR-P 2.0 | National Key Laboratory of Crop Genetic Improvement and Center for Bioinformatics, Huazhong Agricultural University | http://crispr.hzau.edu.cn/cgi-bin/CRISPR2/CRISPR | |
| Software, algorithm | Fiji | Schindelin, J et al., 2012 | https://imagej.net/software/fiji/ | |
| Software, algorithm | Prism | GraphPad | https://www.graphpad.com | |

## Transcriptional profiling of WT and SC-less leaves by RNA-sequencing

Approximately 3-cm long second leaves of WT (Bd21-3) and *sid/bdmute-1* seedlings (7 days after germination grown on ½ MS plates at 20°C with ~100 µmol photons m$^{-2}$ s$^{-1}$ light) were carefully pulled out of the sheath of the first leaf to isolate the leaf developmental zone (3 mm at base of the leaf). 25 developmental zones were collected per replicate and genotype (3 replicates per genotype), snap-frozen in liquid nitrogen and ground using mortar and pistil. Total RNA was isolated using Qiagen's

RNeasy Plant Mini kit with on-column DNAse digestion according to the manufacturer's instructions. The Kapa mRNA HyperPrep (Roche) was used to generate an mRNA enriched sequencing library with an input of 1 µg of total RNA. The libraries were sequenced using the Illumina NextSeq500 platform. Read quality was assessed with FastQC and mapped against the Bd21-3v1.0 genome using bowtie2. Mapped reads were counted using summarised overlap, and differentially expressed genes were analysed using DeSeq2 (*Love et al., 2014*). Finally, gene expression was normalised by transcripts per kilobase million. Raw and processed data are available at Gene Expression Omnibus with the accession number GSE201294.

## Plant material and growth conditions

*B. distachyon* Bd21-3 was used as WT. For plate-based seedling growth, seeds of Bd21-3, mutants, and reporter lines were sterilised for 15 min in 20% bleach (Carl Roth) and 0.1% Triton-100 (Carl Roth), thoroughly rinsed and placed on MS plates (½ MS [Duchefa Biochemie], 1% sucrose [w/v, Carl Roth], 1% bacto agar [w/v, BD], pH 6). The seeds on plates were then stratified and vernalised for at least 2 days at 4°C before transfer to a 28°C chamber with 16 hr light:8 hr dark cycle 110 µmol PAR m$^{-2}$ s$^{-1}$. Plants that were directly transferred to soil, were dehusked, vernalised, and stratified in water for at least 2 days at 4°C and then directly transferred to pots filled with soil, consisting of four parts ED CL73 (Einheitserde) and one part Vermiculite and grown in a greenhouse with 18 hr light:6 hr dark cycle (250–350 µmol PAR m$^{-2}$ s$^{-1}$; day temperature = 28°C, night temperature = 22°C); see also *Haas and Raissig, 2020*.

The three *bdpolar* mutant alleles (*bdpolar-1, bdpolar-2, bdpolar-3*) used for this study were generated by CRISPR/Cas9-based gene editing. The *bdpan1-1* mutant allele was isolated in a forward genetic screen of ethyl methanesulfonate mutagenised Bd21-3 seeds kindly provided by J. P. Vogel (DOE-JGI). All reporter lines in this study were specifically generated (see below).

## Generation of CRISPR/Cas9 lines

Two CRISPR/Cas9 systems were used; the *bdpolar-1* allele was created following the design protocol of *Miao et al., 2013*, while *bdpolar-2* and *bdpolar-3* were generated using the vectors and protocols of *Debernardi et al., 2020*. We used CRISPR-P 2.0 (http://crispr.hzau.edu.cn/cgi-bin/CRISPR2/CRISPR) to select candidate guide RNA spacer sequences and chose suitable guides with a high on-score (few off-score hits). To generate *bdpolar-1*, double-stranded spacer sequences were generated by hybridising the oligo duplexes BdPOLAR-gRNA3-FWD+REV (priMR343 and priMR344) for CRISPR BdPOLAR-gRNA-3 and ligated into the BsaI-HF digested and dephosphorylated pOs-sgRNA vector (*Miao et al., 2013*). Entry clones were then introduced into the destination vector pH-Ubi-Cas9-7 which also encodes the Cas9-enzyme (*Miao et al., 2013*) using LR recombination (Invitrogen). T1 CRISPR/Cas9 lines were genotyped for the absence of the T-DNA construct by using the primers priMR410/411 (Hyg-1F/Hyg-1R). Genotyping of *bdpolar-1* was performed by amplifying the mutated region using priDZ6 and priDZ83 followed by Sanger sequencing of the amplicon. In addition, we designed a CAPS assay for the same amplicon using the BseRI restriction enzyme (Thermofisher); *bdpolar-1* has a+T insertion, which creates a BseRI restriction site in the mutant allele, and only the mutant allele is cut after digestion (*Figure 1—figure supplement 1E*). The *bdpolar-2* and *bdpolar-3* alleles were generated using the protocol from *Debernardi et al., 2020*. Oligo duplexes of *BdPOLAR* guide 2 (priDZ87 and priDZ133 for *bdpolar-2* target site) and guide 4 (priDZ89 and priDZ134 for *bdpolar-3* target site) were hybridised and phosphorylated. Then the oligo duplexes were ligated into the AarI-digested (Thermofisher) destination vector JD633_CRISPRhigheff that encodes for a ubiquitously expressed Cas9 enzyme and regeneration enhancing chimeric GRF–GIF1 protein. To identify the edited mutation, a 665 bp fragment was amplified using priDZ6/priDZ7 including all three target sites of the designed gRNAs and Sanger sequenced.

All primer sequences can be found in *Supplementary file 2*.

## Generation of reporter constructs

Most expression constructs in this study were generated based on the GreenGate system (*Lampropoulos et al., 2013*). Only the *BdPAN1* translational reporter construct was built using the gateway-compatible pIPKb monocot vector series (*Himmelbach et al., 2007*), and the cortical division site marker *BdTAN1* reporter construct was produced via Gibson Assembly.

For GreenGate cloning, we first generated entry modules (pGGA-F). Inserts were amplified with Q5 DNA Polymerase (New England Biolabs) from *Brachypodium* Bd21-3 genomic DNA extracted using the DNeasy Plant Mini Kit (Qiagen) or specific plasmids using primers that contained Type IIS restriction sites and overhangs complementary to the respective entry module. The amplicons were then digested with FastDigest Eco31I (Thermofisher) to reveal the complementary overhangs. Simultaneously, the corresponding backbone (pGGA000 for promoter, pGGC000 for coding sequence, pGGF000 for resistance cassette) was digested using Eco31I and dephosphorylated using antarctic phosphatase (Thermofisher). Then the amplicon was ligated into the backbone using T4 DNA ligase (Thermofisher). All built entry modules were analysed by test digestions, and the inserted fragments were completely Sanger sequenced before proceeding.

### *BdPOLAR* reporter constructs

To produce the entry module pGGA_BdPOLARpro, the 1137 bp intergenic region upstream of *BdPOLAR* was amplified using priDZ2/priDZ30 and ligated into the pGGA backbone. To clone the *BdPOLAR* genomic ORF without STOP into pGGC000, priDZ31 and priDZ35 were designed to mutate and remove the Eco31I site located in the first intron. priDZ3/35 and priDZ31/5 were used to amplify two fragments from Bd21-3 genomic DNA separately, then these two inserts were digested and ligated into digested and dephosphorylated pGGC000 to generate pGGC_BdPOLAR-NS. pGGF_PvUbi2pro-HygR entry module was cloned using switchgrass (*Panicum virgatum*) *ubiquitin 2* (*PvUbi2*) promoter driving hygromycin (Hyg) expression, and the fragment was amplified from pTRANS_250d (*Čermák et al., 2017*). priTN11/priNT30 and priTN12/29 were used to amplify two fragments from pTRANS_250d plasmid separately, then these two inserts were digested and ligated into digested and dephosphorylated pGGF000 to generate pGGF_PvUbi2pro-HygR. To produce the entry module pGGA_ZmUbipro, *ZmUbi* promoter was amplified from pIPK002 (*Himmelbach et al., 2007*) using priTN3/4 and ligated into pGGA000. To generate *BdPOLARp:BdPOLAR-mVenus* expression vector, pGGA_BdPOLARpro, pGGB003 (dummy), pGGC_BdPOLAR-NS, pGGD_linker-mVenus, pGGE001, and pGGF_PvUbi2pro-HygR were introduced into the destination vector pGGZ004 by GreenGate reaction. GreenGate reaction enables the orderly assembly of six entry vectors into the destination vector by a set of seven different overhangs (*Lampropoulos et al., 2013*). It was performed by mixing 1.5 µl of each of the six modules, 1 µl destination vector, 2 µl FastDigest buffer (Thermofisher), 2 µl ATP (10 mM, Thermofisher), 0.5 µl T4 DNA ligase (30 u/µl, Thermofisher), 0.5 µl Eco31I, and 5 µl H$_2$O in a total volume of 20 µl, then incubated for 50 cycles of 37°C for 5 min and 16°C for 5 min each, followed by 50°C for 5 min and 80°C for 5 min. The products of the GreenGate reactions were analysed by test digestions and sequencing of the ligation sites. To clone the expression vector for transcriptional reporter *BdPOLARp:3XNLS-eGFP*, pGGA_BdPOLARpro, pGGB_3xNLS, pGGC_eGFP, pGGD002 (dummy), pGGE001, and pGGF_PvUbi2pro-HygR were assembled into pGGZ004 by GreenGate reaction. To generate C-terminal tagged overexpression *ZmUbip:BdPOLAR-mVenus* reporter, pGGA_ZmUbipro, pGGB003, pGGC_BdPOLAR-NS, pGGD_linker-mVenus, pGGE001, and pGGF_PvUbi2pro-HygR were introduced into pGGZ004 by GreenGate reaction.

The generation of the entry modules pGGA000, pGGB003, pGGC000, pGGD002, pGGE001, and pGGF000 is described in *Lampropoulos et al., 2013*. The entry modules pGGD_linker-mVenus are described in *Waadt et al., 2020*, and pGGZ004 is described in *Lupanga et al., 2020*. The pGGB_3xNLS and pGGC_eGFP were generously provided by Karin Schumacher's group. The pGGD009 (Linker-mCitrine) was generously provided by Jan Lohmann's group.

### *pIPKb_BdPOLAR* genomic construct

Gibson assembly was performed according to the Gibson Assembly Master Mix Kit catalogue (New England Biolabs). priDZ123 and priDZ151 were used to amplify *BdPOLAR* genomic locus including promoter (−1.1 kb) and terminator (+1.9 kb) from Bd21-3 genomic DNA. A total of 0.168 pMol of DNA fragments from *BdPOLAR* were assembled into backbone pIPK001t (*Raissig et al., 2016*). The pIPK001t was digested with OliI and HindIII at 37°C O/N to remove the Gateway cassette and linearise the backbone. The molar ratio (*pIPK001t:BdPOLAR*) used for Gibson reaction was 1:6 in favour of the insert. The products of the Gibson reactions were analysed by colony PCR, test digestion, and Sanger sequencing of the complete insert.

### *BdPAN1* translational reporter construct

First, the BdPAN1 genomic sequence was amplified from Bd21-3 genomic DNA using primers PAN1fwd and PAN1rev and cloned into pENTR/dTOPO (Invitrogen). Then, the clone was digested with AscI (New England Biolabs) and an annealed, double-stranded oligo consisting of Ala_linker-F and Ala_linker-R was ligated in at the C-terminus of the protein. Finally, the resulting construct was re-digested with AscI, and an AscI-flanked Citrine-YFP was ligated in to form the complete construct BdPAN1gene-Alalinker-CitrineYFP pENTR. The Citrine YFP fragment was released by AscI digest from a clone in pJET (Fermentas), which had been originally generated by amplifying the Citrine YFP sequence with primers AscI_FP_noATG-1F and AscI_FP_stop-1R off of plasmid pRSETB-CitrineYFP (*Griesbeck et al., 2001*). Separately, the BdPAN1 promoter was amplified from Bd21-3 genomic DNA using primers PAN1profwd and PAN1prorev and cloned into pCR8/GW/TOPO. It was subsequently digested out using AscI and ligated into AscI-cut destination vector pIPKb001 (*Himmelbach et al., 2007*) to generate BdPAN1pro-pIPKb001. BdPAN1gene-Alalinker-CitrineYFP pENTR was then transferred into this destination vector via LR reaction (Invitrogen) to generate the final construct.

### Cortical division site marker *BdTAN1* reporter

To clone *BdPOLARp:BdTAN1-mCitrine* construct, Gibson assembly was performed according to the Gibson Assembly Master Mix Kit catalogue (New England Biolabs). priDZ123 and priDZ124, priDZ125 and priDZ126 were used to amplify *BdPOLAR* promoter and *BdTAN1* gene from Bd21-3 genomic DNA, respectively. priDZ127 and priDZ128 were used to amplify mCitrine fragments from pGGD009 (Linker-mCitrine). A total of 0.3 pMol of DNA fragments from *BdPOLAR* promoter, *BdTAN1* gene, and mCitrine were assembled into backbone pIPK001t. The pIPK001t was digested with OliI and HindIII at 37°C O/N to remove the Gateway cassette and linearise the backbone. The products of the Gibson reactions were analysed by colony PCR, test digestion, and Sanger sequencing of the whole inserts.

All primer sequences can be found in *Supplementary file 2*.

## Generation and analysis of transgenic lines

*Brachypodium* calli originated from Bd21-3, *bdmute*, Cas9-negative *bdpolar-1*, and *bdpan1-1* plants were transformed with the AGL1 *Agrobacterium tumefaciens* strain combining the protocols of *Alves et al., 2009*; *Bragg et al., 2015*; *Bragg et al., 2012*. In short, young, transparent embryos were isolated and grown for 3 weeks on callus induction media (CIM; per L: 4.43 g Linsmaier & Skoog basal media (LS; Duchefa #L0230), 30 g sucrose, 600 µl CuSO₄ (1 mg/ml, Sigma/Merck #C3036), 500 µl 2,4-D (5 mg/ml in 1 M KOH, Sigma/Merck #D7299), pH 5.8, plus 2.1 g of Phytagel (Sigma/Merck #P8169)). After 3 weeks of incubation at 28°C in the dark, crisp, yellow callus pieces were subcultured to fresh CIM plates and incubated for two more weeks at 28°C in the dark. After 2 weeks, calli were broken down to 2–5 mm small pieces and subcultured for one more week at 28°C in the dark. For transformation, AGL1 Agrobacteria with the desired construct were dissolved in liquid CIM media (same media as above without the phytagel) with freshly added 2,4-D (2.5 µg/ml final conc.), Acetosyringone (200 µM final conc., Sigma/Merck #D134406), and Synperonic PE/F68 (0.1% final conc., Sigma/Merck #81112). The OD600 of the Agrobacteria solution was adjusted to 0.6. Around 100 calli were incubated for at least 10 min in the Agrobacteria solution, dried off on sterile filter paper, and incubated for 3 days at room temperature in the dark. After 3 days, transformed calli were moved to selection media (CIM + Hygromycin [40 µg/ml final conc., Roche #10843555001] + Timentin [200 µg/ml final conc., Ticarcillin 2NA & Clavulanate Potassium from Duchefa #T0190]), and incubated for 1 week at 28°C in the dark. After 1 week, calli were moved to fresh selection plates and incubated for two more weeks at 28°C in the dark. Next, calli were moved to callus regeneration media (per L: 4.43 g of LS, 30 g maltose (Sigma/Merck #M5885), 600 µl CuSO₄ (1 mg/ml), pH 5.8, plus 2.1 g of Phytagel). After autoclaving, cool down and add Timentin (200 µg/ml final conc.), Hygromycin (40 µg/ml final conc.), and sterile Kinetin solution (0.2 µg/ml final conc., Sigma/Merck #K3253). Calli were incubated at 28°C and a 16 hr light:8 hr dark cycle (70–80 µmol PAR m⁻² s⁻¹). After 1–6 weeks in the light, shoots will form. Move shoots that are longer than 1 cm and ideally have two or more leaves, to rooting cups (Duchefa #S1686) containing rooting media (per L: 4.3 g Murashige &Skoog including vitamins [Duchefa #M0222], 30 g sucrose, adjust pH to 5.8, add 2.1 g Phytagel). After autoclaving cool down and add Timentin (200 µg/ml final concentration). Once roots have formed, plantlets can be moved to soil (consisting of four parts ED CL73 (Einheitserde) and one part Vermiculite) and grown in

a greenhouse with 18 hr light:6 hr dark cycle (250–350 µmol PAR m$^{-2}$ s$^{-1}$). Ideally, the transgenic plant-lets moved to soil are initially kept at lower temperatures (day temperature = 22°C, night temperature = 18–20°C) for 2–4 weeks until they have rooted sufficiently before being moved to the warmer greenhouse (day temperature = 28°C, night temperature = 22°C). Gene-edited CRISPR T0 lines were identified by phenotyping and Sanger sequencing of the edited base pairs; reporter T0 lines were verified by examining the signal from fluorescent molecules.

In-depth analysis was done in subsequent generations that went through the seed stage (T1-T3). For *bdpolar* CRISPR mutants, we generated 1 *bdpolar-1*, 1 *bdpolar-2*, and 3 *bdpolar-3* fertile T1 lines, respectively. For *BdPOLAR* reporters from WT, we obtained two translational *BdPOLARp:BdPOLAR-mVenus* T1 lines, three transcriptional *BdPOLARp:3XNLS-eGFP* T1 lines, and five C-terminal tagged overexpression *ZmUbip:BdPOLAR-mVenus* independent T1 lines that were fertile and produced seeds. *BdPOLARp:BdPOLAR-mVenus* in *bdmute* was generated by introducing *BdPOLARp:BdPOLAR-mVenus* into *bdmute* calli. For *BdTAN1* reporters in WT, *bdpolar-1,* and *bdpan1-1,* we obtained five, three, and three fertile T1 lines, respectively.

For all reporter lines either second (5–6 dag at 28°C) or third (8–10 dag at 28°C) leaves of plate-grown plants were used for analysis.

## Crossing

To cross *Brachypodium,* we followed the illustrated guide from John P Vogel (DOE-JGI) (link) with slight modifications. We inspected the plants grown in soil between 4 and 5 weeks and selected those in a proper stage for crossing (florets just before anther dehiscence and with fully developed, feathery stigma). For male parents, 20 or more mature anthers per line were collected on a slide and incubated at 28°C for 10–50 min to induce dehiscence. During the incubation, emasculation was carried out in female parents with fully developed and feathery stigmas. Then pollination was performed by brushing the dehisced anther across the stigma with tweezers. A small piece of tape was then applied at the tip of the flower to ensure that the palea and lemma are tightly closed. 10–30 flowers were pollinated per cross. Reporter lines were always used as pollen donors so signals in the F1 generation indicated a successful cross. In F2, we genotyped the mutant alleles and screened for reporter signals to identify required individuals for analysis. *BdPOLARp:BdPOLAR-mVenus* and *BdPAN1p:BdPAN1-YFP* were crossed as males with *bdpan1-1* and *bdpolar-1* to generate *bdpan1-1;BdPOLARp:BdPOLAR-mVenus* and *bdpolar-1;BdPAN1p:BdPAN1-YFP,* respectively. *BdPOLARp:BdPOLAR-mVenus* and *BdPAN1p:BdPAN1-YFP* were crossed as males with *bdpolar-1* and *bdpan1-1* to generate complementation lines *bdpolar-1;BdPOLARp:BdPOLAR-mVenus* and *bdpan1-1;BdPAN1p:BdPAN1-YFP,* respectively.

## Microscopy and phenotyping

### Light microscopy for phenotyping of mature stomatal complexes

For differential interference contrast (DIC) imaging, the third leaves of soil-grown plants (17~19 dpg) were collected and fixed in 7:1 ethanol:acetic acid overnight to clear the chlorophyll. The cleared leaves were then rinsed in water and mounted on slides in Hoyer's solution (*Liu and Meinke, 1998*). For good optical clearing, leaves were usually incubated for 12–24 hr on the slide before microscopic analysis. The abaxial side was inspected under a ×40 objective using DIC on a Leica DM5000B (Leica Microsystems), and 10–11 fields of view from each individual were saved as pictures for phenotyping SC defects in silico. SCs with transverse or oblique divisions but not correct longitudinal divisions were defined as defective SCs. The numbers of defective SCs and total SCs were counted and recorded simultaneously from each picture. For each defective SC, the frequency of SC divided was counted to distinguish and classify SC division categories (*Figure 4F*). To quantify stomata density, four abaxial fields (0.290 mm$^2$ field of view) per leaf were captured using the ×20 objective. The GC length was measured from the top middle point to the bottom middle point of the GC using the 'straight' tool in Fiji. The stomata density and GC length were quantified using the leaf areas that were used for stomatal conductance measurements in *Figure 5*.

### Confocal microscopy

For confocal imaging, the second or third leaf of plate-grown seedlings (5–6 dpg and 8–10 dpg, respectively) was carefully pulled out to reveal the developmental zone (first 2 mm at the leaf base), which is covered by the sheaths of older leaves. Samples were stained with propidium iodide (PI,

10 μg/ml, Thermofisher) for approximately 5 min to visualise the cell outlines (i.e. the cell walls), then mounted on a slide with water. Imaging was performed using ×63 glycerol immersion objective in Leica TCS SP8 microscope or a Leica Stellaris 5 microscope (Leica Microsystems). For the SP8, we set the Argon laser to 20% and then used the 514 nm laser line (~10–20% intensity) to excite both YFP and PI. For the Stellaris 5, we used 514 nm excitation from a white light LED laser running at 85%. In both cases, two high-sensitivity spectral detectors (HyD) were used to detect fluorescence emission. Images were acquired with a ×63 Glycerol objective at Zoom factor 1, 1024×1024 pixels, and 2–3 line averages. To avoid the variability in localisation patterns due to different focal planes, we always focused on the middle plane for single images after defining top and bottom of each imaged SMC. Also laser intensity for fluorophore excitation was kept constant for a specific line to allow direct comparison unless otherwise noted in the text or figure legends.

Image analysis and processing were carried out in Fiji (*Schindelin et al., 2012*).

## Polarity index
### Polarity Measurement
For POME (*Muroyama et al., 2020*) analysis of BdPAN1-YFP and BdPOLAR-mVenus polarity in WT and mutant backgrounds, confocal images of the developmental zone of second (5~6 dpg) PI-stained leaves were acquired with a ×63 glycerol immersion objective and a ×2 zoom factor. Individual SMCs at stage 3 with strong and clear reporter signals were selected for the analysis. 19–33 SMCs were analysed for each genotype. For the BdPOLAR reporter lines, only cells with signal at the apical, basal, and distal PM were picked. Additionally, recently and incorrectly divided SMCs in both mutant backgrounds were excluded. In some instances where the SMCs were slightly skewed, a small z-stack was obtained and then projected to reveal the complete polarity domain before POME analysis. Fluorescence intensity at each pixel and angle was obtained in ImageJ/FIJI as described by *Muroyama et al., 2020*. Before running POME on a selected cell, the input parameters were specified for each individual cell in the POME FIJI macro. The obtained measurements were then imported, summarised, and analysed in RStudio (details available in *Muroyama et al., 2020*). The polarity index was calculated as the fraction of fluorescence intensity values above the half maximum (*Gong et al., 2021a*). Briefly, the maximum fluorescence intensity of each cell was determined, and the fraction of values above half of this maximum was calculated. The polarity indices of all analysed cells of the same condition were grouped and statistically compared to the other conditions with unpaired Mann-Whitney U-tests.

### Manual tracing
For manual polarity analysis of BdPAN1-YFP, BdPOLAR-mVenus, and BdTAN1-mCitrine polarity in WT, confocal images of the developmental zone of second (5~6 dpg) PI-stained leaves were acquired (midplane, ×63 glycerol immersion objective). Individual SMCs with reporter signal were selected for the analysis (*Figure 2—figure supplement 2*). Fluorescence intensity at each pixel of SMC PM was traced using the Segmented Line tool in Fiji. We always started tracing just above the GMC/SMC interface and completed the whole circumference to obtain the grey value of each pixel in the SMC circumference. The length and width of each neighbouring GMC were also measured using the Straight Line tool (see above 'Nuclear migration assay' for more details). The average fluorescence intensity of BdPAN1-YFP (n=13) and BdPOLAR-mVenus (n=8) each was normalised to their respective average maximum fluorescence intensity and plotted, together with a normalised standard error, displaying the GMC/SMC interface first and then displaying the remaining outline of the SMCs (*Figure 2E*). Average fluorescence intensity of BdTAN1-mCitrine (n=13) and BdPOLAR-mVenus was normalised to their respective average maximum fluorescence intensity. Here, we plotted the intensities of a 4 μm region just before and after the GMC. We did not include the GMC/SMC interface as they vary in length and would have offset the TAN1 and BdPOLAR domains after the GMC.

## Nuclear migration assay
### Staining assay
For nuclear staining experiments, we used the basal 0.5~1 cm developmental zone from second (5–6 dpg) leaves. First, leaf strips were rinsed with 1× PBST three times for 10 min each in vials, then 1 ng/ml Hoechst 33342 (Thermofisher) solution was added into vials, and vacuum was applied three times for 10 min each. Then leaf strips in vials were incubated for 1 hr at room temperature along with gentle

shaking. Afterwards, we washed leaf strips with 1× PBST three times for 10 min each, then added 0.1% Direct red 23 (w/v) (Sigma-Aldrich) into vials, and applied vacuum three times for 10 min each, then incubated leaf strips 1 hr at room temperature along with gentle shaking. Next, we washed leaf strips again with 1× PBST three times for 10 min each. Finally, leaf strips were dried with paper and mounted in 50% Glycerol on slides. Direct red 23 and Hoechst 33342 were used to stain the cell wall and nuclei, respectively. Images were captured with the following excitation (Ex) and emission (Em) wavelengths (Ex/Em): Direct red 23 561 nm/600–650 nm and Hoechst 33342 405 nm/425–475 nm. Image analysis and processing were carried out in Fiji. The d distance, GMC length, and width were measured in Fiji using the 'straight' tool. The d distance was measured starting from the middle point of nuclei in SMC and ending at the middle point of the GMC/SMC interface. The GMC length was measured from the middle point of the apical side to the middle point of the basal side of the GMC. The GMC width was measured from the middle point of the left side to the middle point of the right side of the GMC. To avoid deviations of d distance measurement due to imaging at different Z-axis positions, we also measured the nuclear diameters per genotype, and they did not show a significant difference (*Figure 4—figure supplement 1B*). For quantification of d distance (*Figure 4C*), d distance linear regression in WT (*Figure 4—figure supplement 1A*), and quantification of SMC nuclei diameter (*Figure 4—figure supplement 1B*), each SMC was considered as a replicate. In total 12–30 nuclei of 20–28 individual plants were analysed per genotype; double-blind examination was not performed.

## Multi-parameter phenotyping of cellular defects on simultaneously grown plants

A quantitative experiment on simultaneously plate-grown BdPOLARp:BdTAN1-mCitrine lines in WT (n=19 individuals), *bdpolar-1* (n=23 individuals), and *bdpan1-1* (n=20 individuals) was performed. Plants were initially grown on plates in a growth chamber to the 5-6 dag stage for a collection of immature leaf 2 for SMC analysis, then in a greenhouse to 19 dag after transplanting to the soil, when % abnormal SCs were scored in leaf 3. The second leaf at 5–6 dag was used to measure nuclear migration and BdTAN1 localisation. The nuclear BdTAN1-mCitrine signal was used to quantify the distance d between the SMC nucleus and the GMC/SMC interface, and the LWR of the flanking SMCs was scored to consider only mature SMCs (i.e. flanking GMCs with an LWR >0.9). BdTAN1 localisation was scored in the very same cells of the second leaf. The seedlings of which the second leaf was analysed were then transplanted to soil, and the third leaf was collected at 19 dag, fixed, and cleared. 11–21 fields of view per image at ×40 magnification were imaged and scored for aberrant SC divisions phenotype in the very same individuals that were analysed for the cellular defects of the second leaf. Blind scoring of phenotypes was achieved by having RS acquire and anonymise images and DZ score them.

## Gas exchange measurements

Leaf-level gas exchange measurements were performed as described in *Nunes et al., 2022*. In short, LI-6800 Portable Photosynthesis System (Li-COR Biosciences Inc) was used to measure the youngest, fully expanded leaf (17–21 days after sowing). The LI-6800 was equipped with a Multiphase Flash Fluorometer (6800–01 A) chamber head, and the conditions were as follows: flow rate, 500 µmol s$^{-1}$; fan speed, 10,000 rpm; leaf temperature, 28°C; relative humidity (RH), 40%; [$CO_2$], 400 µmol mol$^{-1}$; photosynthetic active radiation (PAR), 1000–100 – 1000–0 µmol PAR m$^{-2}$ s$^{-1}$ (20 min per light step). Stomatal conductance ($g_{sw}$) measurements were automatically logged every minute. Relative $g_{sw}$ was computed for each individual measured by normalising $g_{sw}$ to the maximum $g_{sw}$ value observed to assess the kinetics of stomatal response by excluding variation in absolute $g_{sw}$. Five individuals each genotype *bdpolar-1*, *bdpan1-1*, *bdpolar-1;bdpan1-1,* and five individual WTs were measured. Stomatal opening and closure speed were evaluated by rate constants (k, min$^{-1}$) determined from exponential equations fitted for each of the three light transitions (1000–100, 100–1000, and 1000–0 PAR), as described in *Nunes et al., 2022*.

## Statistical analysis and plotting

Statistical analysis was performed in R and GraphPad. For multiple comparisons, one-way ANOVA was applied to determine whether the means of all groups are statistically significantly different. Then a Tukey post hoc test was applied to map out which groups are different from other groups. Significant

differences are indicated by different letters. For comparisons between two groups, we used either an unpaired Mann-Whitney U-test or a Welch two-sample t-test as indicated in the figure legends. Significant differences are indicated by asterisks. All plots were done in R using ggplot2 or basic R plotting commands. Plots of stomatal conductance ($g_{sw}$), relative $g_{sw}$, and exponential regressions were done in GraphPad Prism version 9.1.0.

## Acknowledgements

We would like to thank Juliana de Lima Matos and Akhila Bettadapur for help in the original screen that identified the *bdpan1-1* mutant and John P Vogel and the U.S. Department of Energy (DOE) Joint Genome Institute for providing genetic and genomic resources for this original screen. We want to acknowledge the Stanford Functional Genomics Facility for library preparation and next-generation sequencing and Laura R Lee for her support in RNA-seq data analysis. We would like to acknowledge Annika Guse and Jochen Wittbrodt for access to microscopy facilities and Karin Schumacher and Jan Lohmann for providing GreenGate entry modules. Finally, we thank Michael Schilbach for taking care of the greenhouse and our plants, and Upendo Lupanga for technical help and discussions. EA was supported by a U.S. National Science Foundation graduate research fellowship and a Stanford University graduate fellowship. DCB is an investigator of the Howard Hughes Medical Institute. This work was supported by the German Research Foundation (DFG) Emmy Noether Programme Grant RA 3117/1–1 (to MTR).

## Additional information

### Competing interests

Dominique C Bergmann: Reviewing editor, *eLife*. The other authors declare that no competing interests exist.

### Funding

| Funder | Grant reference number | Author |
| --- | --- | --- |
| Deutsche Forschungsgemeinschaft | RA 3117/1-1 | Roxane P Spiegelhalder<br>Inés Hidalgo<br>Barbara Jesenofsky<br>Michael T Raissig<br>Heike Lindner<br>Dan Zhang<br>Tiago DG Nunes |
| National Science Foundation | Graduate research fellowship | Emily B Abrash |
| Howard Hughes Medical Institute | | M Ximena Anleu Gil<br>Dominique C Bergmann |

The funders had no role in study design, data collection and interpretation, or the decision to submit the work for publication.

### Author contributions

Dan Zhang, Formal analysis, Investigation, Visualization, Methodology, Writing - original draft, Writing - review and editing; Roxane P Spiegelhalder, Validation, Investigation, Visualization, Methodology, Writing - review and editing; Emily B Abrash, Inés Hidalgo, Formal analysis, Investigation, Methodology, Writing - review and editing; Tiago DG Nunes, Formal analysis, Investigation, Visualization, Methodology, Writing - review and editing; M Ximena Anleu Gil, Barbara Jesenofsky, Methodology; Heike Lindner, Conceptualization, Formal analysis, Supervision, Investigation, Methodology, Writing - original draft, Writing - review and editing; Dominique C Bergmann, Conceptualization, Supervision, Funding acquisition, Writing - review and editing; Michael T Raissig, Conceptualization, Resources, Formal analysis, Supervision, Funding acquisition, Validation, Investigation, Visualization, Methodology, Writing - original draft, Project administration, Writing - review and editing

### Author ORCIDs

Dan Zhang (ID) http://orcid.org/0000-0003-1385-4817
Roxane P Spiegelhalder (ID) http://orcid.org/0000-0003-2615-1061
Inés Hidalgo (ID) http://orcid.org/0000-0003-4855-446X
Heike Lindner (ID) http://orcid.org/0000-0002-5913-9803
Dominique C Bergmann (ID) http://orcid.org/0000-0003-0873-3543
Michael T Raissig (ID) http://orcid.org/0000-0003-3179-9372

### Decision letter and Author response

Decision letter https://doi.org/10.7554/eLife.79913.sa1
Author response https://doi.org/10.7554/eLife.79913.sa2

---

## Additional files

### Supplementary files

- Supplementary file 1. Differentially expressed genes in *bdmute* developing leaf zones.
- Supplementary file 2. Primers used in this study.
- MDAR checklist

### Data availability

Sequencing data have been deposited in GEO under accession code GSE201294. Source Data files for all figures have been provided.

The following dataset was generated:

| Author(s) | Year | Dataset title | Dataset URL | Database and Identifier |
|---|---|---|---|---|
| Raissig MT, Bergmann DC | 2022 | Comparative RNA-sequencing of Brachypodium distachyon wild-type and bdmute developing leaf zones | https://www.ncbi.nlm.nih.gov/geo/query/acc.cgi?acc=GSE201294 | NCBI Gene Expression Omnibus, GSE201294 |

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
