## [Editor Report]

This study presents a landmark finding that two proteins previously implicated in plant cell polarity, but in different cell types and species, function cooperatively to polarize an asymmetric division in the grass Brachypodium distachyon. The results are well-documented and convincing. The work will be of interest to cell and developmental biologists generally and also to those broadly interested in plant biology.

---

## [Decision Letter]

**Decision letter after peer review:**

Thank you for submitting your article "Opposite polarity programs regulate asymmetric subsidiary cell divisions in grasses" for consideration by *eLife*. Your article has been reviewed by 3 peer reviewers, and the evaluation has been overseen by a Reviewing Editor and Jürgen Kleine-Vehn as the Senior Editor. The following individual involved in the review of your submission has agreed to reveal their identity: Michelle Facette (Reviewer #1).

Essential revisions:

(1) To address the concerns of reviewers 1 and 2 about the lack of clarity regarding the relationship between BdPOLAR and BdPAN1 localization without dual localization of proteins with contrasting FP tags, quantitative analysis is needed to integrate and compare the distribution of BdPOLAR-VENUS vs. BdPAN1-YFP for a population of wild type SMCs (also in the pan1 mutant background, for BdPOLAR-VENUS). A suggestion here is to use data for a single focal plane at a consistent depth (e.g. the midplane), measuring signal intensity along the entire SMC face adjoining GMC + interstomatal cells, for SMCs with a length:width ratio >0.9 where the nucleus is almost always polarized in wild type, but not selected for conformity to the distribution illustrated schematically in Figure 6A. It is further suggested to use plants whose BdPOLAR-VENUS signal intensity is in the "weak" category (as defined for purposes of Figure S4, while explaining in methods or figure legend how that was defined). The rationale for these suggestions is two-fold: (1) Some of the variability in localization patterns seen in the figures may be due to lumping together SMCs at different stages of SMC polarization and/or looking at different focal planes. (2) SMC division defects in BdPOLAR lines with higher expression levels (Figures S4D and perhaps also S4B) may be due to BdPOLAR mislocalization so it makes sense to focus the analysis on lines with low expression. In displaying the results, please indicate the boundaries of the SMC-GMC interface so that signal peaks can be related to these boundaries. Please specify in methods that standardized imaging conditions were used to permit direct comparisons. Note: fulfilling this request will not necessarily require swapping out existing micrograph panels displaying localization findings, however, I concur with the reviewers that Figure 2C and 2F, and the corresponding movies, are more confusing than helpful and that Figure 2G is low priority, so these panels could be replaced with results of quantitative analyses with no loss of impact. The outcome of this quantitative analysis should dictate whether the use of the term "opposite" to describe BdPOLAR vs. BdPAN1 localization patterns (in the title and elsewhere) is appropriate, i.e. if the patterns are perfectly complementary as implied by this term. If not, that's fine but other terminology can be used such as "divergent".

(2) To address similar concerns of reviewers 1 and 2 regarding the lack of clarity in the relationship between BdPOLAR-VENUS vs. BdTAN1-citrine distribution in wild-type plants, quantitative analysis of the same sort requested in (1) is needed to support the conclusion that BdTAN1 rings are located at sites of BdPOLAR depletion, as illustrated schematically in Figure 6.

(3) Clarification is needed to address the concerns of reviewer 1 about the implausible implications of data shown in Figure 4F and the overall picture emerging from the analyses presented in Figures 4B-C and Figure 4E-F. Suggestion: repeat SMC nuclear polarization (Figure 4B-C) and BdTAN1 localization analyses (Figure 4E-F) on a common population of plants grown under the same conditions, whose % defective subsidiaries is also counted and presented, e.g. in a mature portion of the same leaf whose SMC division zone is used for the analysis. If there is a way to visualize nuclei enabling classification of the division stage as part of the TAN1 analysis, that would be ideal but won't be considered essential. Alternatively, the data presented in Figure 4A-F could be dropped from the manuscript, as it wasn't considered essential for key conclusions. If an additional analysis is done and the conclusions hold up, please discuss the seemingly contradictory findings that pan1 mutants have a higher frequency of unpolarized nuclei, yet a lower proportion of SMCs with improperly specified division planes, compared to Bdpolar mutants.

(4) Revise the discussion of "division competence" in light of reviewer 1's point 3.

(5) Please attend to the revisions affecting text or figure composition only (no additional experiments or data analysis needed) as recommended by all 3 reviewers.

*Reviewer #1 (Recommendations for the authors):*

This impactful work should be published, and I don't feel further experiments are necessary. However, I do think the authors should address some inconsistencies in the data, either through rewriting or even potentially leaving some data out.

Figures 1-3 and data therein are beautiful and well controlled. I personally found the top panel in figure S3B more interesting/illustrative than panel G – at the authors' discretion, perhaps consider swapping them?

Re MUTE: "This observation suggested that BdPAN1 polarisation was independent of a successful establishment lateral cell files and indicated that either biochemical or mechanical signals from the elongating GMC polarise BdPAN1." Yes possible (and most likely true) but not the only possibility – MUTE could, in theory, be downstream of PAN1. E.g., the pathway could conceivably be PAN1>MUTE>POLAR, based on available data. I doubt it's true – but it's plausible. (Also, thank you for doing this – I've wanted to know for ages – have you looked at MUTE in pan1 yet?).

Primary concerns:

(1) My biggest concern is with the data in Figure 4F – something doesn't quite "jive" and a lot of inference is drawn from observed differences between pan1 and polar. According to Figure 1, pan1 mutants (in Brachy) have something like ~40% abnormal subsidiaries. If only ~10% of the TAN localization is wrong – that means something goes wrong in Brachy pan1 AFTER the division site is specified – i.e. during phragmoplast guidance. This would be rather shocking, and completely different from maize. It implies that the primary defect in pan1 is not due to polarization, but rather a phragmoplast guidance defect. Coupled with the data that pan1 is (mildly) more affected in nuclear polarization – at least moreso than polar – (as indicated in Figure 4C/Table S5C), yet polar has a more severe defect in TAN localization, the whole thing together leaves me scratching my head.

Ultimately, the data are the data – and since everything is so well controlled I trust it, but I think this has to either be addressed in the text, or careful analysis if some bias was inadvertently introduced. In our lab, we see pretty large variations in % abnormal cells in greenhouse-grown plants. Were the plants within an experiment grown at the same time? Was the quantification done double-blind? How many individual plants (vs cells) were counted – could a single "bad" plant be throwing off the numbers?

If you are confident the numbers are all correct – then I think you must discuss why pan1 mutants have an increased % abnormal subsidiary cells, relatively greater defect in nuclear polarization, yet very little defect in division site specification.

(2) In Figure 4C, n is in the hundreds. I think your p-value might be artificially inflated. I almost wonder if Table S5C is more clear in getting the point across than this scatter plot, which I think indicates that both polar-1 and pan1 have nuclear polarization defects (although the double mutant has a similar defect to pan1. Given that your three alleles of polar have different % defects, and polar-2 is more similar to pan1 polar-1 appears to be a weaker allele, for some unclear reason), I wonder if the difference between pan1 and polar-1 is because you have a weak allele, in which case I would hesitate to interpret them as terribly different.

Coupled with this, I think the assertion that PAN1 governs or guides the nucleus is a bit strong – in many cases the nucleus is just fine. I suggest using "promotes" nuclear polarization. Governs also imply a somewhat direct relationship – which I don't believe there is evidence for here, or in maize.

Putting together 1 and 2 – I honestly feel that a slightly less strong interpretation of nuclear polarization (i.e., that both mutants have a polarization defect) and leaving out figure D-F would have no negative effect on the conclusions, model or impact of the paper. The localization is amazing and interesting! The fact that you know POLAR is dependent on PAN1 is cool! There are future papers to be had here, and I think in future work, (re-)looking a TAN-YFP in conjunction with a microtubule (PPB) marker would clarify things. (Theoretically, you could look at PPBs using immunostaining, but I feel this is beyond the scope necessary here.)

A suggestion, also related to figure 4:

(3) Figure 4G to 4I, regarding the number of aberrant divisions and "cell division capacity".

I think this is useful for you to know, but ultimately, I think your interpretation of the data should be as you want. Acceptance of this manuscript should not be contingent on changing any of this – this is meant to be informative/helpful.

In maize mutants (pan1, pan2, brk1, dcd1, dcd2/o1, etc.) these repeated divisions are quite common, including in mutants I would describe as "late stage" (e.g. o1) where there are no polarity defects. I scoured past papers and realized that we don't really explicitly talk about it much (and perhaps even inadvertently avoided images showing it), so mea culpa. The fact that SMCs divide once, and only once, is an important and interesting point (indeed – it makes this model system unique from many other asymmetric in that it is NOT a stem cell division), and carefully documenting it is useful. We can even see SMCs that are in the process of re-dividing after the oriented division of the GMC. I think the most likely explanation for repeated rounds of division is when the division plane is incorrect, subsidiary fate is not specified – perhaps due to incorrect segregation (or concentration of?) known or unknown fate factors. Note that incorrectly divided subsidiaries, fate is sometimes, and sometimes not, correctly specified (see Gallagher and Smith, 2000).

I can see why you hypothesized that POLAR contributes to "capacity", based on the known roles of AtPOLAR. I am unconvinced from the data that polar and pan1 are meaningfully different in this regard – in both cases multiple divisions are happening many times (as opposed to the near zero times in wildtype). This is especially true since you only have a single allele for each mutant, and again, it is unclear how many different plants (as opposed to cells) are analyzed. Coupled with the head-scratcher in 4F – I might not speculate to the extent that these data suggest unique functions for PAN1 and POLAR. The extrapolation that POLAR might be uniquely related to division capacity seems like a big stretch. On the other hand, these data link the only discussion of how At and Bd POLAR might have opposing functions, so it might be nice to keep some discussion, in conjunction with an alternate hypothesis. Either way, I think this is the author's call, but wanted to share these observations.

Other notes: I think something that would be lovely to have done is to make PAN1 and POLAR localization constructs in different colors, so they could be imaged at the same time. I think it is NOT worth holding up publication for this – but I spent a long time looking at Figure 2 trying to figure out how the PAN1 "stripes" overlap (or don't) with POLAR. It would also help resolve the relative timing of the two proteins. Similarly, I tried to determine if POLAR lines up with where the PPB/division zone is. POLAR seems to be excluded from a discrete domain that seems to align more closely with the PPB than PAN1 in the "shoulders" and I don't know if that is consistent or just in the images in Figure 2F. In particular, a microtubule marker to determine how the division site aligns (or does not) would be useful. Since PAN1 is pretty far away from the division site, how or why.

Finally – I think the abstract gives short shrift to how you discovered POLAR (transcriptional profiling of mute) and I think, based on my critique, overemphasizes "distinct roles" for POLAR and PAN1.

*Reviewer #2 (Recommendations for the authors):*

1. "ABdPAN1p:BdPAN1-YFP reporter line was expressed…at the cell periphery (Figure 2A; Figure S3A)." Delete "line".

2. Equilateral triangular arrowheads are used to point to the objects in Figures 2E and 3E. However, it is difficult to identify the points indicated by these marks. Replace them with irregular arrowheads as seen in Figure 4E.

3. "Just after cell division, BdPOLAR quickly dissociated from the distal PMs (Figure 2E; Figure S3B)." Is this the correct figure citation? I could not recognize BdPOLAR dissociating from PMs in Figure 2E.

*Reviewer #3 (Recommendations for the authors):*

I thoroughly enjoyed reading this manuscript and am particularly excited about your findings regarding BdPOLAR polar localization and function.

---

## [Author Response]

Essential revisions:(1) To address the concerns of reviewers 1 and 2 about the lack of clarity regarding the relationship between BdPOLAR and BdPAN1 localization without dual localization of proteins with contrasting FP tags, quantitative analysis is needed to integrate and compare the distribution of BdPOLAR-VENUS vs. BdPAN1-YFP for a population of wild type SMCs (also in the pan1 mutant background, for BdPOLAR-VENUS). A suggestion here is to use data for a single focal plane at a consistent depth (e.g. the midplane), measuring signal intensity along the entire SMC face adjoining GMC + interstomatal cells, for SMCs with a length:width ratio >0.9 where the nucleus is almost always polarized in wild type, but not selected for conformity to the distribution illustrated schematically in Figure 6A. It is further suggested to use plants whose BdPOLAR-VENUS signal intensity is in the "weak" category (as defined for purposes of Figure S4, while explaining in methods or figure legend how that was defined). The rationale for these suggestions is two-fold: (1) Some of the variability in localization patterns seen in the figures may be due to lumping together SMCs at different stages of SMC polarization and/or looking at different focal planes. (2) SMC division defects in BdPOLAR lines with higher expression levels (Figures S4D and perhaps also S4B) may be due to BdPOLAR mislocalization so it makes sense to focus the analysis on lines with low expression. In displaying the results, please indicate the boundaries of the SMC-GMC interface so that signal peaks can be related to these boundaries. Please specify in methods that standardized imaging conditions were used to permit direct comparisons. Note: fulfilling this request will not necessarily require swapping out existing micrograph panels displaying localization findings, however, I concur with the reviewers that Figure 2C and 2F, and the corresponding movies, are more confusing than helpful and that Figure 2G is low priority, so these panels could be replaced with results of quantitative analyses with no loss of impact. The outcome of this quantitative analysis should dictate whether the use of the term "opposite" to describe BdPOLAR vs. BdPAN1 localization patterns (in the title and elsewhere) is appropriate, i.e. if the patterns are perfectly complementary as implied by this term. If not, that's fine but other terminology can be used such as "divergent".

To address these concerns, we have undertaken several efforts. First of all, we provide more details regarding image acquisition to visualize polarity domains (p. 20). Indeed, we always imaged the midplane of SMCs, which we determine by identifying the top and bottom of each SMC. Also, laser intensity was kept constant within a line to allow direct comparison, except where we specifically mention that this was not the case.

Second, we manually traced the outlines of SMCs expressing BdPAN1-YFP (LWR >0.8) or “weak” BdPOLAR-YFP (LWR >0.9) to determine where polarity domains reside relative to one another (see new *Figure 2—figure supplement 2* for an overview of the analyzed cells). We have used SMCs with a LWR >0.8 for BdPAN1-YFP cells, as BdPAN1 is solidly polarized much earlier than BdPOLAR. We plotted the average fluorescence intensity of the two markers in the same plot, always displaying the GMC/SMC interface first and then displaying the remaining outline of the SMCs. This plot and a schematic model of what it is representing are now displayed as *Figure 2E* replacing Figure 2C and 2F. The data becomes quite noisy towards the end as SMCs tend to have different sizes (and thus different cell circumferences), but it is obvious that BdPOLAR is the lowest and BdPAN1 is the highest at the GMC interface. While it is not perfectly complementary (i.e. there is a part of the SMC where neither BdPAN1 nor BdPOLAR are detected), we still would strongly argue for these polarity domains to be opposite. Imaging conditions were standardized within a maker line but are slightly different between marker lines (i.e. laser intensity) as they are expressed at different levels. We have therefore normalized the average fluorescence intensity levels of each reporter for direct comparison (relative average fluorescence intensity was plotted). We have included a paragraph regarding this analysis in the results part on p. 6 and added detailed methods on p. 21.

We agree that analysis of both BdPAN1 and BdPOLAR reporters in the same cells would be an excellent addition, and to this end, we generated a dual marker line (BdPAN1p:BdPAN1-CFP; BdPOLARp:BdPOLAR-mCitrine). However, we found that the BdPAN1-CFP signal was too weak compared to the mCitrine signal to be able to visualize the proximal BdPAN1 domain. It is a lengthy process and technical challenge to create dual reporter lines in grasses, and because of this disappointing dual-reporter result, we instead focused on the quantitative analysis of reporter distribution detailed in the previous paragraph.

(2) To address similar concerns of reviewers 1 and 2 regarding the lack of clarity in the relationship between BdPOLAR-VENUS vs. BdTAN1-citrine distribution in wild-type plants, quantitative analysis of the same sort requested in (1) is needed to support the conclusion that BdTAN1 rings are located at sites of BdPOLAR depletion, as illustrated schematically in Figure 6.

As requested we have generated a very similar analysis of BdPOLAR-YFP and BdTAN1-mCitrine. We re-used the data from above for BdPOLAR-YFP (same as Figure 2E) and in addition, manually traced the circumference of BdTAN1-mCitrine lines. We again normalized the average fluorescence intensity for each line and plotted the normalized fluorescence intensity of the 4µm before and after the GMC/SMC interface. We did not plot the signal along the GMC/SMC interfaces as these varied in size and would have created variable TAN1 peaks and POLAR gaps after the GMC/SMC interface. The plot and one representative image are now shown as new Figure 4G-I. This figure replaces previous panels (Figure 4G-I, division capacity control) which we moved to the supplementary information to address point 4 and the criticism about division competence (see below). We added a few sentences explaining our observations in the result part (p. 9) and provided details regarding the analysis in the methods (p. 21)

(3) Clarification is needed to address the concerns of reviewer 1 about the implausible implications of data shown in Figure 4F and the overall picture emerging from the analyses presented in Figures 4B-C and Figure 4E-F. Suggestion: repeat SMC nuclear polarization (Figure 4B-C) and BdTAN1 localization analyses (Figure 4E-F) on a common population of plants grown under the same conditions, whose % defective subsidiaries is also counted and presented, e.g. in a mature portion of the same leaf whose SMC division zone is used for the analysis. If there is a way to visualize nuclei enabling classification of the division stage as part of the TAN1 analysis, that would be ideal but won't be considered essential. Alternatively, the data presented in Figure 4A-F could be dropped from the manuscript, as it wasn't considered essential for key conclusions. If an additional analysis is done and the conclusions hold up, please discuss the seemingly contradictory findings that pan1 mutants have a higher frequency of unpolarized nuclei, yet a lower proportion of SMCs with improperly specified division planes, compared to Bdpolar mutants.

We would like to thank reviewer 1 for pointing this out. Indeed, it is unexpected that we observe so few cortical division site defects (i.e. BdTAN1 localization) when observing relatively frequent SC division defects. Particularly in *bdpan1,* where nuclei are not polarized normally, we get <10% BdTAN1 localization defects but ~40% of aberrant SCs. We agree that this is in stark contrast to the recent observations in maize (Ashraf et al. (2022) bioRxiv).

Again, we have undertaken several efforts to address these seemingly conflicting observations. First and as requested, we have repeated the quantification of nuclear migration defects, BdTAN1 localization and mature SC division defects on a set of simultaneously grown BdTAN1-mCitrine reporter lines in WT, *bdpolar* and *bdpan1* backgrounds. Importantly, the person that took the images (RS) anonymized the images and the first author DZ then analyzed the randomized images. We used the weak nuclear BdTAN1-mCitrine signal to quantify the distance d between SMC nucleus and the GMC/SMC interface (again considering LWR of the flanking SMCs), and have scored the BdTAN1 localization in the very same cells of the second leaf. We then collected the third leaf at 19 dag and scored the mature SC division phenotype in these very same individuals. Not only could we confirm the results in Figure 4, we now only observe nuclear migration defects in *bdpan1* and not in *bdpolar* and find a similar frequency of mature SC division defects in *bdpan1* and *bdpolar*. We have included the new data in *Figure 4—figure supplement 1E, F and G,* added a paragraph in the results part to address the new experiment (p. 9) and summarized the methods in detail (p. 22).

Second, we have addressed these unexpected observations in a new paragraph in the discussion part on p. 12. We address that indeed polarity defects in *B. distachyon* SMCs could affect phragmoplast-related processes. In addition, we offer a hypothesis that considers the cell geometry of SMCs, which seems to differ between *B. distachyon* and maize; in particular, SMCs in *B. distachyon* are less wide relative to their length and usually do not present distinctly skewed cell walls just above and below the GMC/SMC interface. We, therefore, speculate that the extending phragmoplast is more likely to short-cut to the distal wall in *B. distachyon* SMCs when SMC polarity is compromised.

(4) Revise the discussion of "division competence" in light of reviewer 1's point 3.

We want to thank Prof. Facette for openly sharing her findings that this seems to be a common observation in many of her SMC polarity and ACD mutants. We, therefore, have moved the quantification of SMC division frequencies to *Figure 4—figure supplement 2* (in favor of BdTAN1-YFP and BdPOLAR colocalization data, see above) and adapted the results (p. 9-10) and discussion (p. 13) accordingly.

(5) Please attend to the revisions affecting text or figure composition only (no additional experiments or data analysis needed) as recommended by all 3 reviewers.

We have done so, please see details below.

Reviewer #1 (Recommendations for the authors):This impactful work should be published, and I don't feel further experiments are necessary. However, I do think the authors should address some inconsistencies in the data, either through rewriting or even potentially leaving some data out.Figures 1-3 and data therein are beautiful and well controlled. I personally found the top panel in figure S3B more interesting/illustrative than panel G – at the authors' discretion, perhaps consider swapping them?Re MUTE: "This observation suggested that BdPAN1 polarisation was independent of a successful establishment lateral cell files and indicated that either biochemical or mechanical signals from the elongating GMC polarise BdPAN1." Yes possible (and most likely true) but not the only possibility – MUTE could, in theory, be downstream of PAN1. E.g., the pathway could conceivably be PAN1>MUTE>POLAR, based on available data. I doubt it's true – but it's plausible. (Also, thank you for doing this – I've wanted to know for ages – have you looked at MUTE in pan1 yet?).

This is an interesting idea and we have not looked at *MUTE* in *pan1*. We have, though, added a sentence to the discussion/results on p. 6 to pay tribute to this scenario. Owing to this comment, we have recently transformed the YFP-MUTE reporter into *bdpan1* to address this, but the lines are not yet ready for analysis.

Primary concerns:(1) My biggest concern is with the data in Figure 4F – something doesn't quite "jive" and a lot of inference is drawn from observed differences between pan1 and polar. According to Figure 1, pan1 mutants (in Brachy) have something like ~40% abnormal subsidiaries. If only ~10% of the TAN localization is wrong – that means something goes wrong in Brachy pan1 AFTER the division site is specified – i.e. during phragmoplast guidance. This would be rather shocking, and completely different from maize. It implies that the primary defect in pan1 is not due to polarization, but rather a phragmoplast guidance defect. Coupled with the data that pan1 is (mildly) more affected in nuclear polarization – at least moreso than polar – (as indicated in Figure 4C/Table S5C), yet polar has a more severe defect in TAN localization, the whole thing together leaves me scratching my head.Ultimately, the data are the data – and since everything is so well controlled I trust it, but I think this has to either be addressed in the text, or careful analysis if some bias was inadvertently introduced. In our lab, we see pretty large variations in % abnormal cells in greenhouse-grown plants. Were the plants within an experiment grown at the same time? Was the quantification done double-blind? How many individual plants (vs cells) were counted – could a single "bad" plant be throwing off the numbers?If you are confident the numbers are all correct – then I think you must discuss why pan1 mutants have an increased % abnormal subsidiary cells, relatively greater defect in nuclear polarization, yet very little defect in division site specification.

This is a fair criticism and we would like to thank the reviewer for giving us an incentive to investigate and discuss this more extensively.

To address this criticism and the request of the reviewing editor, we have repeated the quantification of nuclear migration defects, BdTAN1 localization and mature SC division defects on a set of simultaneously grown BdTAN1-mCitrine reporter lines in WT, *bdpolar* and *bdpan1*. For details regarding methods, findings and new discussion, please refer to Essential Revisions #3 above.

(2) In Figure 4C, n is in the hundreds. I think your p-value might be artificially inflated. I almost wonder if Table S5C is more clear in getting the point across than this scatter plot, which I think indicates that both polar-1 and pan1 have nuclear polarization defects (although the double mutant has a similar defect to pan1. Given that your three alleles of polar have different % defects, and polar-2 is more similar to pan1 polar-1 appears to be a weaker allele, for some unclear reason), I wonder if the difference between pan1 and polar-1 is because you have a weak allele, in which case I would hesitate to interpret them as terribly different.Coupled with this, I think the assertion that PAN1 governs or guides the nucleus is a bit strong – in many cases the nucleus is just fine. I suggest using "promotes" nuclear polarization. Governs also imply a somewhat direct relationship – which I don't believe there is evidence for here, or in maize.

The new, simultaneous experiment actually does not find a nuclear polarization defect in *bdpolar* anymore. Thus, this indeed indicates that there is a different effect on nuclear migration in *bdpolar* and *bdpan1*. Nonetheless, we toned down the conclusion and use “promote” rather than “govern” throughout the text.

Regarding weak and strong alleles, we feel that all three *bdpolar* alleles are actually strong alleles. They all have a single nucleotide insertion fairly early in the gene creating a non-sense peptide sequence after the modification (and preliminary STOPs in all three cases). The *bdpan1* allele, however, might be a rather weak allele since it has an in-frame 9nt deletion and thus merely misses 3 amino acids. So if we underestimate a mutant effect, then this is more likely the case for the *bdpan1* rather than the *bdpolar* mutant. Yet, we agree that the SC division defect frequency is fairly variable as implicated by variable frequencies in Figure 1D, Figure 3—figure supplement 1 and Figure 4—figure supplement 1.

Putting together 1 and 2 – I honestly feel that a slightly less strong interpretation of nuclear polarization (i.e., that both mutants have a polarization defect) and leaving out figure D-F would have no negative effect on the conclusions, model or impact of the paper. The localization is amazing and interesting! The fact that you know POLAR is dependent on PAN1 is cool! There are future papers to be had here, and I think in future work, (re-)looking a TAN-YFP in conjunction with a microtubule (PPB) marker would clarify things. (Theoretically, you could look at PPBs using immunostaining, but I feel this is beyond the scope necessary here.)

As suggested and mentioned above we have toned down the interpretation. Nonetheless, the repeated, “simultaneous, double-blind” experiment supported our previous findings and this is why we think it is appropriate to retain the data in Figure 4D-E.

Regarding immunostainings and/or microtubule markers to visualize PPB, the developmental zone in *B. distachyon* is much smaller than in maize and there are only a handful of SMCs that polarize and divide per leaf (hence the high number of individuals (often >20) required for these analyses). Therefore, visualizing the highly transient PPB using immunostainings is very hard in our hands (we tried), which is why we decided to use the mitotically stable TAN1 marker instead.

A suggestion, also related to figure 4:(3) Figure 4G to 4I, regarding the number of aberrant divisions and "cell division capacity".I think this is useful for you to know, but ultimately, I think your interpretation of the data should be as you want. Acceptance of this manuscript should not be contingent on changing any of this – this is meant to be informative/helpful.In maize mutants (pan1, pan2, brk1, dcd1, dcd2/o1, etc.) these repeated divisions are quite common, including in mutants I would describe as "late stage" (e.g. o1) where there are no polarity defects. I scoured past papers and realized that we don't really explicitly talk about it much (and perhaps even inadvertently avoided images showing it), so mea culpa. The fact that SMCs divide once, and only once, is an important and interesting point (indeed – it makes this model system unique from many other asymmetric in that it is NOT a stem cell division), and carefully documenting it is useful. We can even see SMCs that are in the process of re-dividing after the oriented division of the GMC. I think the most likely explanation for repeated rounds of division is when the division plane is incorrect, subsidiary fate is not specified – perhaps due to incorrect segregation (or concentration of?) known or unknown fate factors. Note that incorrectly divided subsidiaries, fate is sometimes, and sometimes not, correctly specified (see Gallagher and Smith, 2000).I can see why you hypothesized that POLAR contributes to "capacity", based on the known roles of AtPOLAR. I am unconvinced from the data that polar and pan1 are meaningfully different in this regard – in both cases multiple divisions are happening many times (as opposed to the near zero times in wildtype). This is especially true since you only have a single allele for each mutant, and again, it is unclear how many different plants (as opposed to cells) are analyzed. Coupled with the head-scratcher in 4F – I might not speculate to the extent that these data suggest unique functions for PAN1 and POLAR. The extrapolation that POLAR might be uniquely related to division capacity seems like a big stretch. On the other hand, these data link the only discussion of how At and Bd POLAR might have opposing functions, so it might be nice to keep some discussion, in conjunction with an alternate hypothesis. Either way, I think this is the author's call, but wanted to share these observations.

We would like to thank Prof. Facette for this carefully argued, very valid comment and for sharing that in the maize mutants these repeated divisions are common yet nowhere explicitly stated. We have indeed looked at this knowing that AtPOLAR controls division capacity in Arabidopsis. We have moved Figure 4G-I into supplemental materials and now discuss this observation still in the light of AtPOLAR’s function, but not implying a unique role for BdPOLAR compared to BdPAN1.

In addition, we would like to mention that (at least in *B. distachyon* and other Pooid systems (Stebbins and Shah, 1960)) some SMCs flank two GMCs and asymmetrically divide twice (which make SMCs kind of a stem cell). Yet how this is regulated is unclear. It could be that there is a signal from GMCs that keep division competency high.

Other notes: I think something that would be lovely to have done is to make PAN1 and POLAR localization constructs in different colors, so they could be imaged at the same time. I think it is NOT worth holding up publication for this – but I spent a long time looking at Figure 2 trying to figure out how the PAN1 "stripes" overlap (or don't) with POLAR. It would also help resolve the relative timing of the two proteins. Similarly, I tried to determine if POLAR lines up with where the PPB/division zone is. POLAR seems to be excluded from a discrete domain that seems to align more closely with the PPB than PAN1 in the "shoulders" and I don't know if that is consistent or just in the images in Figure 2F. In particular, a microtubule marker to determine how the division site aligns (or does not) would be useful. Since PAN1 is pretty far away from the division site, how or why.

As mentioned above, we have tried generating a dual marker line (BdPAN1p:BdPAN1-CFP; BdPOLARp:BdPOLAR-mCitrine), yet unfortunately, the BdPAN1-CFP signal (compared to mCitrine signal) was too weak to visualize the proximal BdPAN1 domain in these lines.

To determine how BdPOLAR and BdPAN1 relate spatially to each other, we have added data in *Figure 2E* where we manually traced mature SMC outlines to determine BdPOLAR-mVenus and BdPAN1-mCitrine occupancy along the SMC’s circumference. This confirmed that the polarization is indeed opposite yet not perfectly reciprocal (see details above, Essential Revisions #1). We did the same thing for BdTAN1-mCitrine and plotted relative average fluorescence intensity in *Figure 4G-I* nicely showing that BdTAN1 indeed resides in the BdPOLAR gaps above and below the GMC (again, details above, Essential Revisions #2).

Finally – I think the abstract gives short shrift to how you discovered POLAR (transcriptional profiling of mute) and I think, based on my critique, overemphasizes "distinct roles" for POLAR and PAN1.

We have extended the abstract to highlight how we discovered BdPOLAR and toned down “distinct roles” to “diverse/various roles” as both proteins might contribute to each other's function on a cellular level.

Reviewer #2 (Recommendations for the authors):1. "ABdPAN1p:BdPAN1-YFP reporter line was expressed…at the cell periphery (Figure 2A; Figure S3A)." Delete "line".

Done.

2. Equilateral triangular arrowheads are used to point to the objects in Figures 2E and 3E. However, it is difficult to identify the points indicated by these marks. Replace them with irregular arrowheads as seen in Figure 4E.

Done, thank you for the suggestion.

3. "Just after cell division, BdPOLAR quickly dissociated from the distal PMs (Figure 2E; Figure S3B)." Is this the correct figure citation? I could not recognize BdPOLAR dissociating from PMs in Figure 2E.

Thank you for pointing out this. Indeed, we have changed Figure 2E and it is not easily visible anymore now. We have highlighted the cell where BdPOLAR is dissociated post-division with an arrow and changed the legend accordingly.